# Trade-offs between multifunctionality and profit in tropical smallholder landscapes

Ingo Grass ⬤ et al.#

Land-use transitions can enhance the livelihoods of smallholder farmers but potential economic-ecological trade-offs remain poorly understood. Here, we present an inter-disciplinary study of the environmental, social and economic consequences of land-use transitions in a tropical smallholder landscape on Sumatra, Indonesia. We find widespread biodiversity-profit trade-offs resulting from land-use transitions from forest and agroforestry systems to rubber and oil palm monocultures, for 26,894 aboveground and belowground species and whole-ecosystem multidiversity. Despite variation between ecosystem functions, profit gains come at the expense of ecosystem multifunctionality, indicating far-reaching ecosystem deterioration. We identify landscape compositions that can mitigate trade-offs under optimal land-use allocation but also show that intensive monocultures always lead to higher profits. These findings suggest that, to reduce losses in biodiversity and ecosystem functioning, changes in economic incentive structures through well-designed policies are urgently needed.

---

#A full list of authors and their affiliations appears at the end of the paper.

Agricultural expansion and intensification are the main threats to tropical biodiversity and ecosystem functioning[1,2]. In recent years, increasing attention has been paid to the rise of large-scale commercial agriculture in the tropics, and particularly, to controversial crops such as oil palm. However, it is smallholders with farms smaller than 5 ha who manage the largest share of agricultural land in many tropical regions, even among so-called "estate crops" such as oil palm and rubber[3,4]. Although smallholders strongly shape tropical landscapes, consequences of their land-use choices for socioeconomic and ecological functions remain poorly studied[5,6]. Tropical smallholder landscapes are typically characterized by a mosaic of forest fragments, agroforestry, monocultures, and settlements, which hold the potential to combine high yields and high biodiversity[7,8]. Instead, many tropical landscapes are undergoing widespread land-use transitions, with smallholders shifting from traditional low-input systems to intensively managed and more-profitable monocultures[9]. These transitions are likely to result in economic–ecological trade-offs, with economic profit increasing from intensified land-use at the expense of tropical biodiversity and ecosystem functioning[10,11]. However, economic functions are rarely directly related to ecological outcomes, although the shape of their relationships, such as whether trade-offs are linear or non-linear, has important management implications.

The UN Sustainable Development Goals (SDGs) aim at managing landscapes for improved livelihoods while ensuring the conservation and sustainable use of terrestrial ecosystems[12]. Despite being fundamental toward achieving the SDGs, our understanding of recent tropical land-use transitions as driving forces of economic–ecological trade-offs remains limited. Research is particularly scarce when it comes to assessing whole-ecosystem biodiversity (multidiversity[13]) and ecosystem functioning (multifunctionality[14]) across multiple tropical land uses in different transitional stages, which is, however, a prerequisite for successful planning of future tropical landscapes in light of the SDGs.

Here, we explore trade-offs and synergies between multidiversity (26,894 species across 14 taxonomic groups), multifunctionality (36 indicators of 10 ecosystem functions), and profitability (annual profits per hectare after deducting production costs from revenues) across multiple land uses with a significant share of smallholders in Jambi Province on the island of Sumatra, Indonesia. In contrast to previous interdisciplinary work in our study system[10,15], we explicitly model relationships between profits and ecological functions to ascertain the shape of profit–function relationships. We investigate economic–ecological trade-offs for an, to the best of our knowledge, unprecedented number of taxonomic groups and ecosystem functions, as well as with indices of multidiversity and multifunctionality that characterize the whole-ecosystem state of land-use systems. Finally, we aim to scale-up from plot to landscape scale by identifying optimized landscape compositions that mitigate trade-offs between ecological functions and rising profit expectations from smallholder land use.

Our study region is both a global biodiversity hotspot and a showcase of ongoing agricultural expansion by formal (i.e., transmigration until the 1990s) and informal (i.e., occupation) land-use transitions: between 1990 and 2013, rainforest land cover in Jambi Province decreased from 49.5% to 34.5%, whereas the land under rubber and oil palm cultivation increased from 26.4% to 32.5% (Fig. 1). Losses in rainforest cover primarily amounted to transformation to rubber and oil palm plantations, other agricultural land uses, and shrub, i.e., land after deforestation that is usually converted to plantations after few years of fallow (Fig. 1; Supplementary Table 1). By 2017, 99% of the land under rubber and ~61% of the land under oil palm in Jambi was cultivated by smallholders[16]. Moreover, jungle rubber, a traditional agroforestry system of rubber-enriched disturbed or secondary forests, which was formerly the main rubber production system in the region, has become economically marginal owing to its low returns to land and labor (Supplementary Table 2), resulting in its conversion to more-profitable rubber and oil palm monocultures[10]. To understand the economic–ecological trade-offs of these smallholder land-use transitions from lowland rainforest to jungle rubber agroforestry and intensive rubber and oil palm monocultures, we (1) conduct extensive ecological and socioeconomic field studies, (2) including continuous records of rubber and oil palm yields in the same study plots over a 2-year period, and (3) derive yield–profit relationships based on the management practices of 700 smallholder farm households in our study region. We focus on profits as these can be expressed per unit of land, and profits are positively associated with other economic and human welfare dimensions in our study system, such as household incomes, food security, and consumption expenditures of smallholders[17,18]. Furthermore, to resolve spatial landscape planning and the underlying political drivers of land-use transformation, we conduct 150 stakeholder interviews with government representatives, NGOs and corporate actors at district, provincial, and national level. We find that smallholder land-use transitions from forest and agroforestry systems to rubber and oil palm monocultures generally result in substantial economic–ecological trade-offs. Increases in profits of farmers occur at the cost of massive losses in biodiversity and of key ecosystem functions, indicating far-reaching ecosystem deterioration. Although some trade-offs may be mitigated under optimal land-use allocation, our findings question the long-term sustainability of ongoing economic development in this global biodiversity hotspot. Changes in economic incentive structures through well-designed policies are urgently needed.

## Results and discussion

**Biodiversity-profit trade-offs.** In our biodiversity assessments, we used an extensive sample of tropical biodiversity comprising a total of 26,894 species and operational taxonomic units, including the most diverse and highest biomass groups and all trophic levels[19]. We found strong evidence for non-linear losses in species richness with increasing profits from smallholder land use across the majority of taxonomic groups (Fig. 2a). Losses were generally most pronounced at the transition from forest and jungle rubber to monocultures, with the former two land-use systems showing the poorest profitability but the highest species richness (Fig. 2b). By contrast, by generating incomes of up to 1000 USD ha$^{-1}$ year$^{-1}$, rubber and particularly oil palm monocultures were significantly more profitable (Fig. 2b); however, they generally harbored the lowest levels of biodiversity. Although the total species richness of a few groups was not related to or even increased with higher profits (e.g., bacteria), negative richness-profitability relationships were pervasive when analyzing the subset of species that also occurred in rainforest (47% of all species; Fig. 2a). Hence, although more-profitable monocultures may partially support biodiversity by species turnover (i.e., replacement of rainforest species with habitat generalists or exotic species), rainforest transformation to monoculture plantations negatively affected rainforest species across all taxonomic groups. These findings are particularly noteworthy because of our focus on smallholder plantations that are typically much less intensively managed than large, commercial estates: oil palm smallholders use on average only half the amount of nitrogen and phosphorus that is applied in oil palm estates[20]. Win–win situations of profit increases without reducing biodiversity—as reported from cocoa agroforestry[21]—are therefore not evident when smallholders shift from traditional but less-profitable

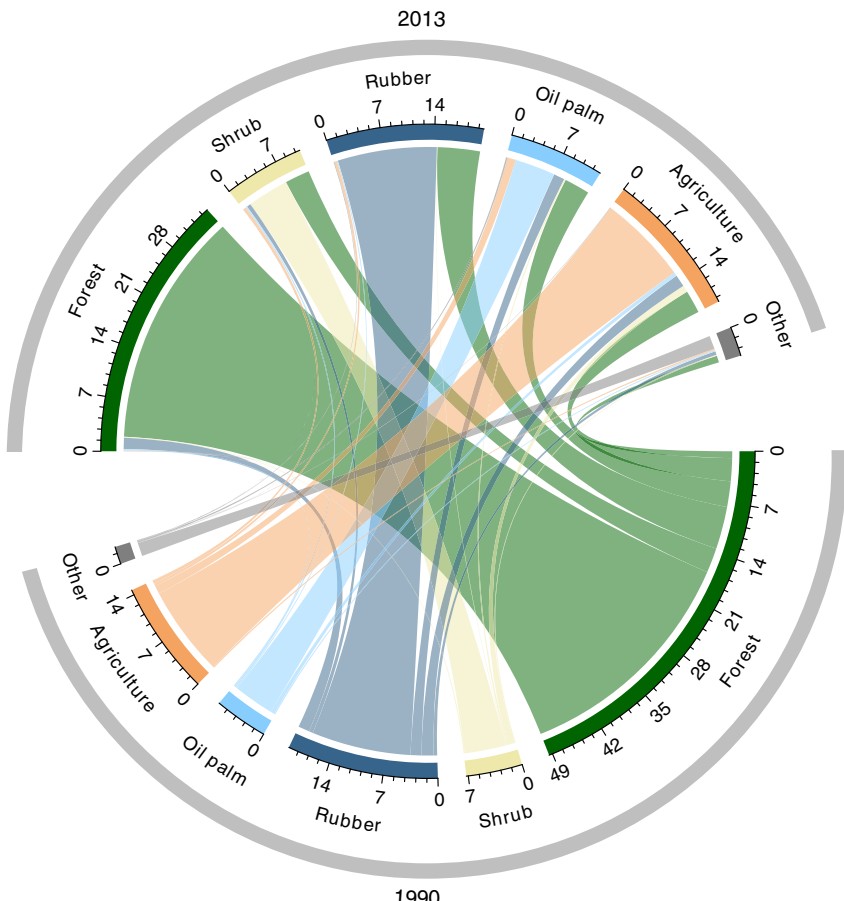

**Fig. 1 Land cover changes and agricultural transitions (%) from 1990 to 2013 in Jambi Province (Sumatra, Indonesia).** Note that rubber contains both land cover by jungle rubber and rubber monoculture plantations, as these could not be clearly distinguished by remote sensing. Colored flows show pathways of land-use transitions from 1990 to 2013, excluding intermediary steps (e.g., transitions from rainforest to shrub to oil palm).

agroforestry to more intensively managed monocultures. Moreover, trade-offs are pervasive for both aboveground and belowground biota.

**Ecosystem function-profit trade-offs.** In addition, we studied relationships between profit and ecosystem functioning for 36 indicators representing 10 ecosystem functions (Fig. 3). The majority of relationships indicated undesirable trade-offs: key functions such as soil and aboveground carbon stocks, soil respiration as an indicator of belowground biotic activities, and decomposition declined, whereas nutrient leaching and greenhouse gas fluxes increased with higher profits from land use (Fig. 3)[22,23]. Indicators of soil fertility improved with increasing profits, but only because soil amendments (lime, borate, and phosphorus fertilizers) were applied to the oil palm plantations in these inherently acidic Acrisol soils (Fig. 3)[23]. Some relationships between ecosystem function indicators and profit were hump-shaped or U-shaped (i.e., plant transpiration, climatic conditions; Fig. 3), indicating complex system-specific responses to land-use transition[24,25].

**Whole-ecosystem multidiversity and multifunctionality.** We then calculated multidiversity[13] and multifunctionality[14] that, respectively, comprise all taxa and ecosystem functions for each plot, in order to test for trade-offs between increasing profits from land-use transitions and whole-ecosystem biodiversity and functioning. These indices are commonly calculated as the proportion of plot-level measured taxonomic groups or functions of which

performance exceeds an a priori minimum defined threshold (e.g., 70%, 50%, or 30%) of their maximum measured performance level[14]. The maximum performance level is thereby not restricted to a specific land-use system; for instance, some functions may peak in rainforest plots, whereas others may peak in monoculture plantations. Since defining a specific threshold that determines whether a given group or function contributes to multidiversity or multifunctionality can be arbitrary, we calculated the full range of thresholds from 1% to 99%[14,26]. This approach also allowed investigating whether relationships with profitability differed with stringency of land-use management for multidiversity and multifunctionality: management expectations of multifunctionality based on a 90% threshold are much more stringent than calculations based on a 50% threshold, for example. We found clear trade-offs between multidiversity and land-use profitability, which were observed for the entire threshold range (Fig. 4). Moreover, we observed a consistent loss of multifunctionality with higher profits across the full threshold range (Fig. 4). Trade-offs for both multidiversity and multifunctionality were strongest for thresholds approximately within the 30–70% range (Fig. 4). Increasing profitability of land use thus always comes at the expense of the overall ecosystem diversity and functioning, even when land-use management aims at retaining only medium to low levels of multidiversity or multifunctionality.

**Land-use composition to mitigate trade-offs.** Finally, we asked if and under which constraints it is possible to design tropical landscapes that maintain biodiversity and ecosystem functioning,

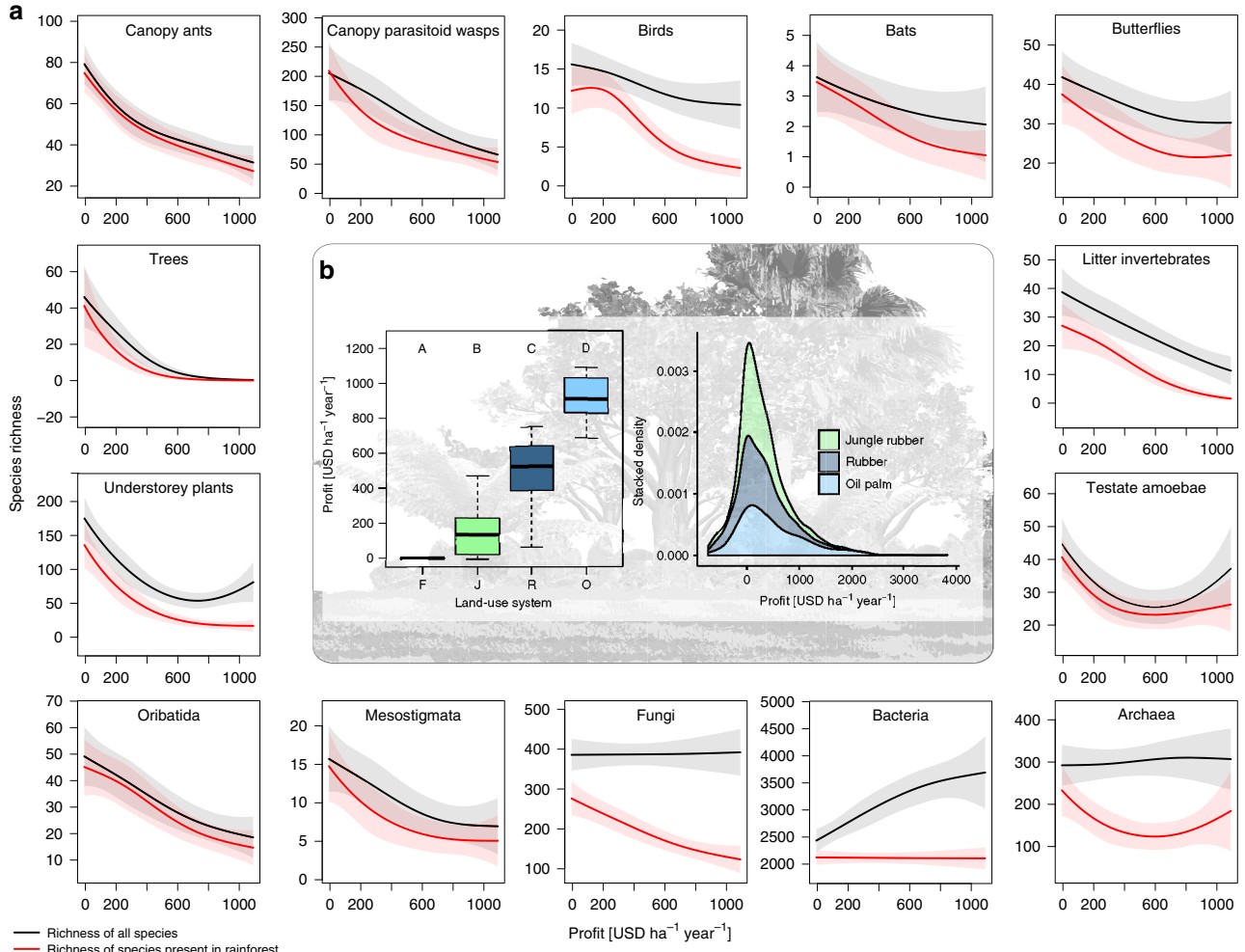

**Fig. 2 Species richness changes non-linearly with increasing profits from land-use transitions by smallholders in an Indonesian landscape.** Land-use systems were primary degraded lowland rainforest (F), agroforestry jungle rubber (J), rubber monoculture (R), and oil palm monoculture (O). **a** Species richness and profit estimates were derived from plot-level data in eight replicates per land-use system. Biodiversity-profit trade-offs were predicted using simulation-extrapolation (SIMEX) of richness-profit relationships (thin lines: SIMEX predictions; shaded areas: 95% confidence bands). Predictions for biodiversity were based on species richness of all species (black lines) and of species that were present in rainforest (red lines). **b** Mean profit per land-use system based on the crop yields in the 32 ecological study plots (left panel) and Kernel density estimates of profit distributions from 701 smallholder household interviews (right panel). Boxplots represent the median (black bars), the 25–75% intervals (box edges) and the 1.5 interquartile range (whiskers). Letters indicate significant differences in profits of land-use systems (Tukey test, $P < 0.05$). Source data are provided as a Source Data file.

and yet allow increases in profits from smallholder land use. To this end, we used a genetic algorithm to generate conceptual landscapes with optimized land-use composition for maximum multidiversity/multifunctionality with increasing profit demands from agricultural production ranging from 0 to 1000 United States dollars (USD) average profit per ha and year. Each conceptual landscape consisted of 32 empty slots (referring to the 32 study plots from the ecological field studies) to be filled by the genetic algorithm with the plot-level data from our field surveys. Filling was done with replacement, i.e., the algorithm could select the same plot(s) multiple times and thereby exclude others from selection. Hence, the total search space covered ~$9.16 \times 10^{17}$ possible combinations. The simulated landscapes with optimized composition for a given profit expectation converge along the production-possibility frontier, i.e., the Pareto-frontier for balancing economic–ecological trade-offs at the landscape scale. In other words, the Pareto-frontier provided a set of multiple optimum landscape compositions, which cannot be further optimized (e.g., by higher biodiversity) under the given constraints (i.e., the minimum expected profits per ha). Our simulations indicated

that maintaining high levels of multidiversity required high proportions of lowland rainforest at the landscape scale, regardless of whether multidiversity was calculated based on all species or only species also present in rainforest (Fig. 5; Supplementary Fig. 1). With higher profit demands, trade-offs became unavoidable, and the replacement of rainforest with plantations resulted in parallel and linear decreases of rainforest cover and multidiversity (Fig. 5). All medium (400–600 USD ha$^{-1}$ year$^{-1}$) to highly profitable (>800 USD ha$^{-1}$ year$^{-1}$) landscapes were dominated by oil palm plantations (Fig. 5; Supplementary Fig. 1), suggesting that despite the considerable loss of multidiversity, the trade-off would have been even stronger if profits had been primarily derived from jungle rubber or rubber plantations instead. The high importance of rainforest in maintaining landscape scale diversity was further supported by separate optimizations for each of the 14 taxonomic groups, with highly similar landscape compositions across aboveground and belowground taxa (Supplementary Fig. 2). Generally, a mixture of different land uses often resulted in the highest biodiversity at the landscape scale (Supplementary Fig. 2), emphasizing the importance of species

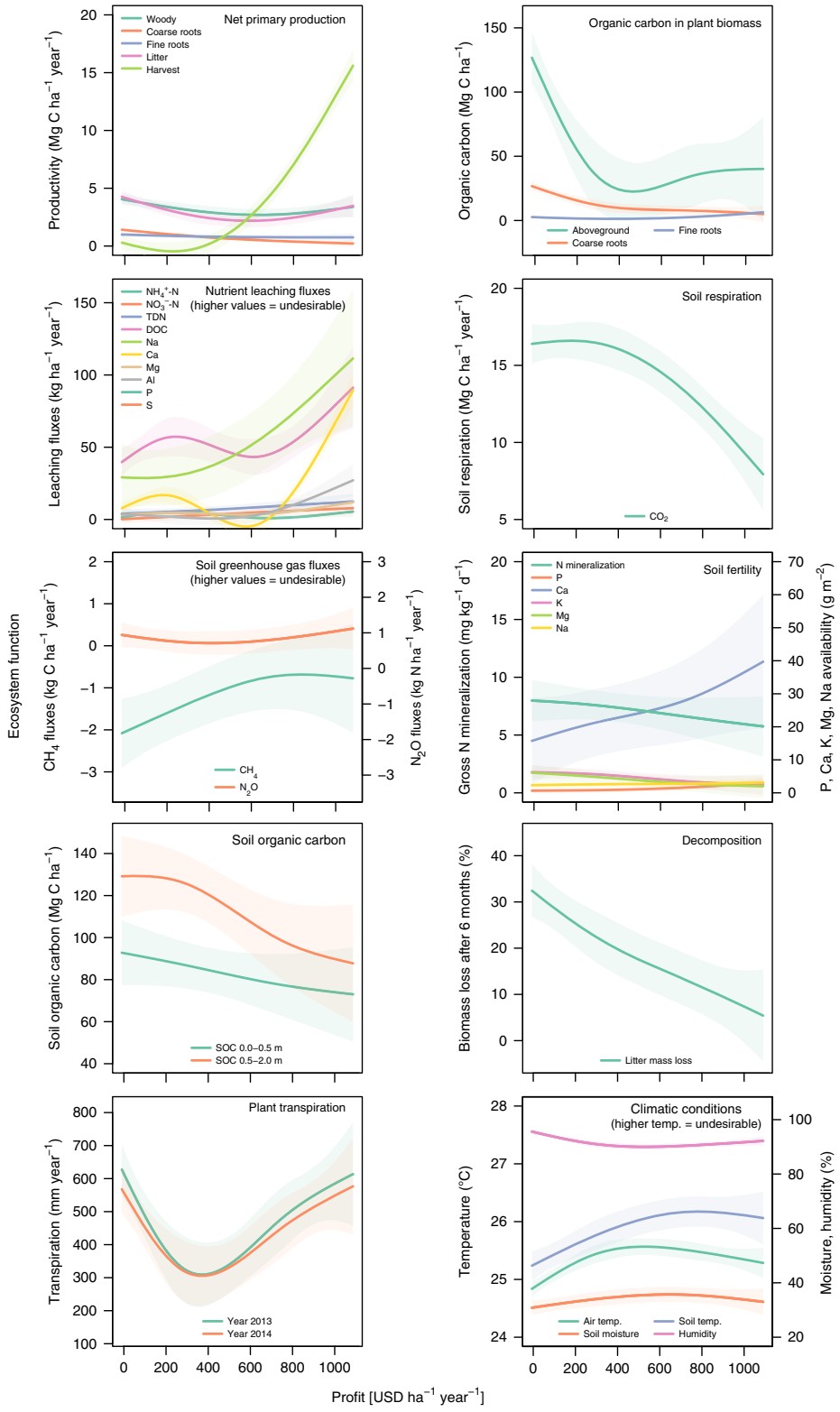

**Fig. 3 Ecosystem functions and their indicators show diverging responses to increasing profits from land-use transitions by smallholders in an Indonesian landscape.** Shown are responses of function indicators (thin lines: SIMEX predictions; shaded areas: 95% confidence bands). Source data are provided as a Source Data file.

turnover between land uses for biodiversity conservation[27]. Analogous to the biodiversity simulations, we generated optimized landscapes for each of the 10 ecosystem functions and multifunctionality. We found their compositions were highly contingent on the targeted ecosystem function and

profit expectation (Fig. 5; Supplementary Fig. 3). Landscapes designed to maintain high levels of soil respiration or low levels of nutrient-leaching fluxes were generally dominated by rubber plantations. By contrast, the algorithm included oil palm plantations in these landscapes only when profits exceeding

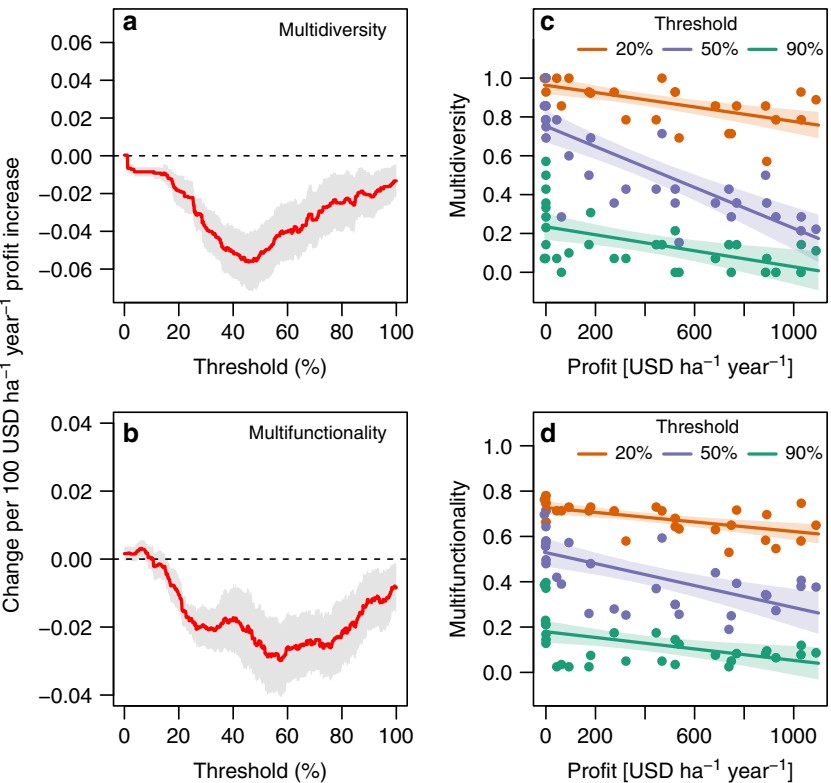

**Fig. 4 Multidiversity-profit and multifunctionality-profit relationships are generally negative, regardless of thresholds used to define multidiversity or multifunctionality.** Changes in **a** multidiversity (whole-system biodiversity of all species in 14 taxonomic groups) and **b** multifunctionality (whole-system ecosystem functioning based on 36 indicators of 10 functions) per profit increase of 100 USD ha$^{-1}$ year$^{-1}$ in smallholder farms. All indices range between 0 and 1, whereby 1 is the highest-possible level of multidiversity or multifunctionality. Note that relationships are usually negative, regardless of the threshold that taxonomic groups or ecosystem functions need to reach to contribute to multidiversity or multifunctionality (red lines are predicted slopes; shaded areas indicate 95% confidence bands). In addition, shown are examples of **c** multidiversity-profit and **d** multifunctionality-profit relationships for thresholds of 20%, 50%, and 90%, respectively (points are raw data; fitted lines and shaded areas are predictions from linear models and 95% confidence bands, respectively; $P < 0.05$ in all cases, tested with simple linear regression). Source data are provided as a Source Data file.

800 USD ha$^{-1}$ year$^{-1}$ were expected, which, however, entailed strong function losses (Supplementary Fig. 3). For other functions, e.g., NPP or soil greenhouse gas fluxes, oil palm plantations caused less trade-offs than rubber systems (Supplementary Fig. 3). Moreover, we found that not all functions could be sustained at similar levels. For example, given optimal land-use allocation, high levels of soil greenhouse gas fluxes could be avoided even for landscapes with average profits exceeding 800 USD ha$^{-1}$ year$^{-1}$. Contrarily, given similar profit expectations, decomposition and organic carbon storage retained <50% of their maximum potential (Fig. 5; Supplementary Fig. 3). The diverging responses of functions to landscape composition resulted in low multifunctionality even at low-profit expectations and regardless of thresholds used in the calculations (Supplementary Fig. 1). Moreover, multifunctionality decreased linearly from poorly to highly profitable landscapes (Fig. 5). In summary, these results suggest that, whereas mitigating biodiversity-profit trade-offs under land scarcity is most efficiently achieved by retaining rainforest habitat and deriving most profits from oil palm monocultures, there is no one-size-fits-all solution for maintaining high levels of ecosystem multifunctionality with increasing profits from smallholder agriculture in these tropical landscapes.

**Implications.** We found that higher profits from agricultural transitions in Indonesia's tropical smallholder landscapes occur at the cost of massive biodiversity losses and deterioration of terrestrial ecosystems. These findings question the long-term

sustainability of ongoing economic development in this global biodiversity hotspot, and showcase threats from tropical land-use transitions worldwide[28]. The loss of ecosystem multifunctionality demonstrated in this study not only affects local livelihoods but also has far-reaching effects beyond, including the loss of soil and aboveground carbon stocks and concurrent greenhouse gas emissions that accelerate global climate change[29,30]. Moreover, although transition to more-profitable land uses can improve living standards for the better-off, these transitions are not possible for those households that lack adequate knowledge or access to land and resources, especially non-farm households. Concurrently, our in-depth interviews with rural farm and non-farm households revealed increasing social inequality and land tenure conflicts with oil palm expansion in the region[17,31].

The concrete results reported here are specific for Jambi Province. However, although some of the details may differ by region, the general findings on the economic–ecological trade-offs will likely also hold for other parts of Indonesia and tropical lowland regions worldwide. The economic impacts of the oil palm boom for smallholders reach far beyond those assessed here in the profit function. Indeed, adoption of oil palm production has not only increased household incomes, but also enhanced food security, nutrition, and consumption expenditures of adopting smallholder farmers in Jambi Province[17,18,32]. At national level, it is estimated that the oil palm boom since 2000 may have lifted up to 2.6 million rural Indonesians out of poverty[33]. However, our study shows that more-targeted landscape planning is needed to increase land-use efficiency and ensure social and ecological

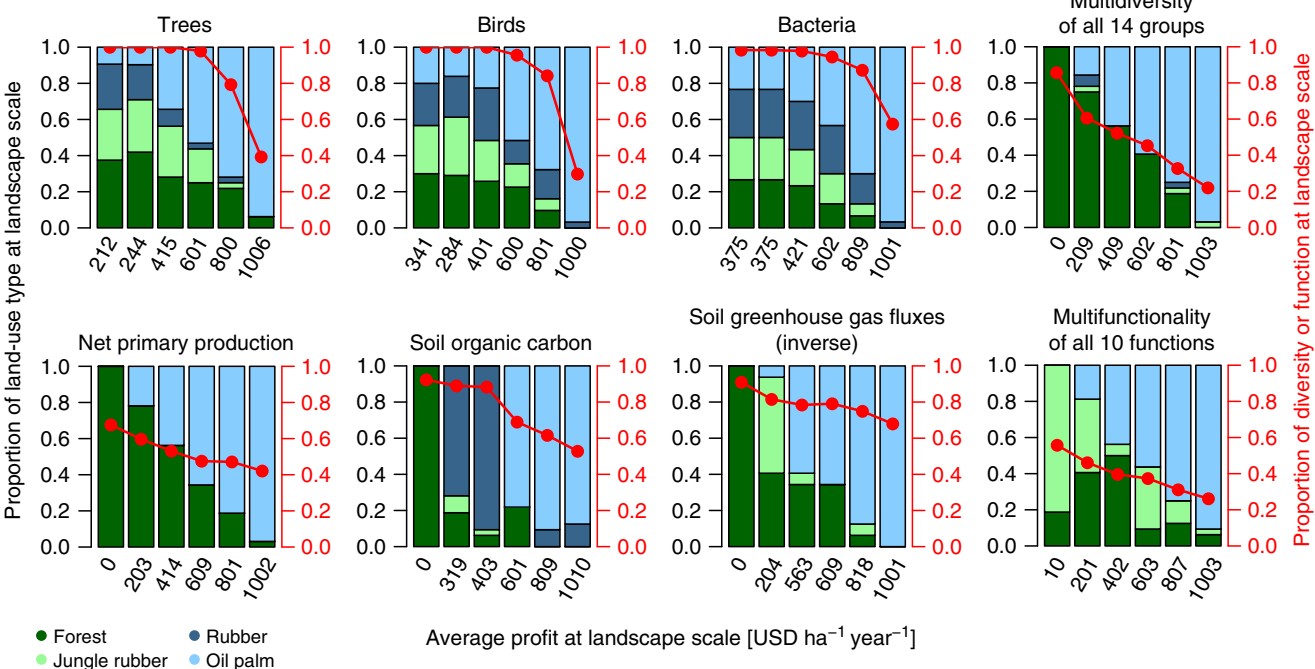

**Fig. 5 Optimized landscapes for highest-possible levels of biodiversity or ecosystem functioning with increasing profits from agricultural production.** Optimized landscape compositions are shown for examples of taxonomic groups and ecosystem functions as well as for multidiversity and multifunctionality considering all studied groups and functions, respectively. Each bar represents a landscape solution as identified by a genetic algorithm, fed with plot-level information on biodiversity or ecosystem functions and profits of smallholder farmers. Colors indicate the composition of landscape solutions, i.e., the proportional share of the four studied land-use systems. Red dots indicate the realized biodiversity or ecosystem function for a given landscape composition, connected by lines to visualize trends with increasing profit expectations. Realized values are scaled between 0 and 1, whereby 1 corresponds to 100% of biodiversity (all sampled species present) or ecosystem functioning (all function indicators at their maximum) at the landscape level. A priori defined profit expectations: 0; 200; 400; 600; 800; 1000 USD ha$^{-1}$ year$^{-1}$. Source data are provided as a Source Data file.

sustainability. In particular, multifunctional landscapes in the tropical lowlands require context-specific solutions that overcome profit-functioning trade-offs that remain unavoidable without changing the economic incentives for smallholders. A combination of well-designed regulatory command and control measures with incentive-based measures such as payment for environmental services (PES) schemes is one promising option[34]. Premium prices for outputs produced with ecologically friendly practices, such as rubber or palm oil from certified landscapes that include production and non-production land, are an alternative[35]. Any approach will require law enforcement and the consideration of trade-offs between multifunctionality and profit in spatial planning to halt unsustainable land-use change and biodiversity loss in tropical lowlands.

## Methods

**Study region and study design.** Field research for this study was carried out in the tropical lowlands of Jambi Province on the island of Sumatra, Indonesia. This region has experienced massive land-use change and transition over the course of the 20th century and is a showcase of smallholder agriculture in Indonesia[10]. For much of the last century, the region was characterized by shifting cultivation and subsistence farming, but with the introduction of transmigration villages in the 1980s under the Suharto regime, the underlying political development strategy focused on market-oriented modern crops and smallholder expansion supported by transmigration[36]. Contract farmers for palm oil production made up the vast majority of transmigrants that were moved from the densely-populated island of Java to Jambi Province[36]. The rise of plantation agriculture resulted in transformation and loss of primary rainforest, which still continued in the 21st century[10]. In 2011, the Indonesian government presented its ambitious master plan to accelerate nation-wide economic development until 2025, which also includes the Sumatra Economic Corridor, a large-scale infrastructure and development project to further transform the island into Indonesia's mainstay of plantation agriculture[37]. Within Jambi Province, we selected two landscapes, "Harapan" and "Bukit Duabelas" (with loam and Acrisol soils, respectively) with four land-use systems common to the region:

primary degraded lowland rainforest[38], jungle rubber (a traditional agroforestry system), monoculture rubber plantation and monoculture oil palm plantation. At the time of site selection in 2012, the monoculture rubber plantation varied between 7 and 16 years in age and the oil palm plantations between 8 and 15 years. In each landscape, we established four 50 m × 50 m replicate plots in each of the four land-use systems, resulting in 32 study plots. This number of replicates is typical for ecological studies in tropical lowland forests[39]. Forest plots were located in the Bukit Duabelas National Park and the Harapan Rainforest Restoration concession (PT REKI). All other plots were owned and managed by smallholders. Within each plot, five permanent 5 m × 5 m subplots were established. More information on the study region, study design, land-use systems, and management practices of the smallholder systems are published elsewhere[10,15].

**Quantification and mapping of land use and land cover changes from 1990 to 2013 in Jambi Province.** Figures for land cover and land-use change were derived from a spatio-temporal model based on official geodata available for the years 1990 and 2013. The land-use classifications were produced by the Indonesian Ministry of Environment and Forestry following a standard methodology based on Landsat and SPOT imagery[40]. Although the official land cover map distinguishes 23 land cover classes (seven forest classes, 15 non-forest classes, one class of clouds/no data), they were aggregated and refined into the five main classes relevant for this study[41]: forest (including primary and secondary rainforest as well as forest plantations), oil palm, rubber (containing rubber plantations and jungle rubber), other agricultural systems and shrub/bush land. A change matrix of aggregated land-use classes between the years 1990 and 2013 was computed by intersecting all single polygons and deriving the related area that had changed from one class to another. An independent validation of the existing maps was only possible based on collected field data for the map product from 2013. The accuracy assessment showed an overall accuracy of classification of 82.6% and Kappa coefficient of 0.79. In general, each land use had >70% accuracy with a relative balance between producer and user accuracy. In absence of an independent validation for the map from 1990, we here need to assume a comparable accuracy. Accuracies for the respective map products reported in ref. [40] are higher (88% for all 23 classes), but refer to whole Indonesia and not to Jambi Province alone.

**Yield assessment.** From beginning of July 2014 to end of June 2016, especially trained field assistants and plot managers continuously monitored the harvested

biomass (tapped and dried rubber, fresh oil palm bunches) from the selected 50 × 50 m jungle rubber, monoculture rubber, and monoculture oil palm plots. Following their normal schedule, plantation keepers tapped rubber trees continuously and collected rubber ranging from every week to twice per month. Oil palm bunches were harvested on average twice per month. Yields were standardized to rubber and fresh fruit bunch weight in kg ha$^{-1}$ and averaged across years.

**Trees and understorey vegetation**. Within each plot, we identified all trees with a diameter at breast height ≥ 10 cm to species level. Moreover, all vascular plant individuals growing within the five 5 m × 5 m subplots were identified, including terrestrial plants (herbs, shrubs, and young trees), climbers, and epiphytes up to 3 m height[42]. Herbarium specimens of three individuals per species were prepared for identification and later deposition at several Indonesian herbaria (Herbarium Bogoriense, BIOTROP Herbarium, Herbarium of the University of Jambi, Harapan Rainforest Herbarium).

**Canopy ants and parasitoid wasps**. Canopy ants (Formicidae) and canopy parasitoid wasps (Braconidae, Ceraphronidae, Encyrtidae, Eulophidae, Platygastridae, Scelionidae) were collected by canopy fogging. Per fogging event, we used 50 ml DECIS 25 (Bayer Crop Science, active ingredient: Deltamethrine 25 g/L) dissolved in 4 L of petroleum white oil, applied to target canopies by the Swingtec SwingFog SN50. All 32 study plots were sampled twice, first in the dry season 2013 and the second time in the rainy season 2013/2014. Standardization was achieved by placing 16 funnel traps (1 m$^2$ each) underneath each target canopy in three subplot replicates per plot[15]. In total, we sampled 130,527 individuals of canopy ants, sorted to 227 (morpho-)species from nine subfamilies (dry season 2013 and rainy season 2013/2014) and 10,070 individuals of parasitoid wasps, sorted to 1,182 morphospecies (dry season 2013).

**Birds and bats**. Birds were sampled with point counts as well as automated sound recordings from May to July 2013. All plots were visited three times for 20 min point counts. The observer was in the plot middle, and all birds detected within the plot were recorded. Point counts took place between 6:00 and 10:00 and the timing for individual plots alternated between early and late morning. We excluded detections from fly-overs, and bird vocalizations that could not be identified immediately were recorded using a directional microphone (Sennheiser ME-66) to compare with recordings from the Xeno-Canto online bird call database (http://xeno-canto.org/). In addition to point counts, we recorded stereo sound at 44,100 Hz sampling frequency (SMX-II microphones, SM2+ recorder, Wildlife acoustics); the recorders were attached to the central tree of the plot at 2.0–2.5 m height. We could record sound in eight plots simultaneously; sampling all 32 plots took four days (10th and 13th of May, and the 3rd and 7th of June 2013). We uploaded the first 20 min after sunrise to a website (http://soundefforts.uni-goettingen.de/) so that two independent ornithologists could identify all audible and visible bird calls (within an estimated 35 m radius) to species. For each plot, only bird species identified by both ornithologists were subsequently merged with the species obtained from the point counts to generate the data set used in the analysis.

Bats were caught using mist nets and harp traps between April and August 2015. We used telescopic aluminum poles to install mist nets with a total of 48 m in length as well as two harp traps (1.35 m × 1.75 m) in presumed bat flyways. Mist nets were 3 m high, with 19 mm in mesh size, and installed at ground level or up to 3 m high, depending on plot conditions. Each plot was sampled from 17:30 to 22:00 on two consecutive nights for an average sampling effort of 1296 m$^2$ of mist net hours. We checked the nets and traps every 15 min until 20:00 and every 30 min thereafter. Harp traps were left on site and checked the morning after. Each individual bat's morphology was measured to identify them according to the latest bat species checklist for Sumatra[43]. We tagged the bats' nails with nail polish color codes to identify recaptures and released them after closing mist nets.

**Butterflies**. We obtained abundance data for butterflies (Lepidoptera: Papilionidae, Pieridae, Lycaenidae, Nymphalidae) from all 32 study plots between August and October 2017. Butterflies were collected using sweep netting (exception: *Troides amphrysus* CRAMER 1779, identified on sight) on three parallel transects per plot, with two transects located on the outer borders of the plots, and the third transect located through the center. Sweep netting was conducted twice per day per plot, in the morning (8:00–11:00 am) and afternoon hours (13:00–16:00 pm). All butterfly individuals were released after identification in the evenings of the sampling day, with the exception of up to two dried/mounted individuals and five individuals in 99% EtOH p.a. per species, which were kept for species ID and further analysis. Our data are based on 6653 caught and/or observed butterfly individuals that we identified to 209 species, using standard taxonomic literature.

**Litter invertebrates**. We sampled litter macroinvertebrates in three subplots of each of the 32 study plots between October and November 2012[44,45]. In each subplot, we sieved 1 m$^2$ of leaf litter from the ground through a 2 cm width mesh and hand-collected all invertebrates visible to the naked eye from the containers below the sieves. Animals were stored in 65% ethanol for further identification in the laboratory. All animal individuals were then identified to family and subsequently, given a lack of suitable identification keys for the study area, to

morphospecies based on consistent morphological characteristics. Juvenile spiders were excluded from the data set, as they could not be reliably identified to morphospecies. Finally, observed litter invertebrate species richness was calculated as the number of morphospecies present in the total 3 m$^2$ sampled at each study plot.

**Testate amoebae**. To sample testate amoebae (protists) at each study plot, we took litter and upper mineral soil samples (to a depth of 50 mm) in October and November 2013, using a corer of 50 mm in diameter[46]. We then extracted testate amoebae from the samples by washing 1 g dry weight litter sample over a filter of 500 μm mesh and back-sieving the filtrate through 10 μm mesh. Microscopic slides were prepared from the final filtrate and testate amoebae were identified to morphospecies[46].

**Oribatida and mesostigmata**. To collect soil microarthropods (Oribatida and Mesostigmata) three soil cores were taken during October to November 2013 from each study plot. Soil cores measured 16 cm × 16 cm and comprised the litter layer and the underlying mineral soil layer to a depth of 5 cm. Animals were extracted by heat[47], collected in dimethyleneglycol-water solution (1:1) and thereafter transferred into 70% ethanol. More details on the sampling and extraction procedure are given in ref. [48]. Oribatida were identified to (morpho)species from two out of three cores in each plot, Mesostigmata were identified to (morpho)species from one out of three cores in each plot.

**Fungi**. In each plot, we collected soil samples in three subplots resulting in 96 samples. After sieving, freeze-drying and storage at −20 °C under liquid nitrogen[49], DNA was isolated with the PowerSoil DNA Isolation Kit (MO BIO Laboratories Inc.), reverse transcribed, amplified with the ITS1-F-KYO1 and ITS-4 primers and linked with 454 pyrosequencing adaptors (Roche, Mannheim, Germany). Purified products were submitted to the Göttingen Genomics Laboratory (G2L, Göttingen, Germany) for sequencing and bioinformatics analyses including sequence assembly and quality filtering[50]. For taxonomic assignment, high-quality sequences were blasted against the UNITE database (v7, sh_refs_qiime_ver7_99_s_01.08.2015.fasta); unclassified OTUs and extrinsic domain OTUs (Protista, Plantae) were removed[50]. Sequences were deposited in the National Center for Biotechnology Information (NCBI) Sequence Read Archive (SRA) under accession number SRP134264. A rarefied OTU table (1229 sequences per sample) was used for the current analyses.

**Bacteria and archaea**. To assess bacterial and archaeal community compositions bulk soil DNA of three subplots per study plot was extracted with PowerSoil DNA isolation kit (Dianova, Hamburg, Germany) and used for amplification of 16 S rRNA genes targeting the V3–V5 region using the Phusion hot start high-fidelity DNA Polymerase (Finnzymes)[51]. The thermal cycling scheme was as follows: initial denaturation at 98 °C for 5 min, 25 cycles of denaturation at 98 °C for 45 s, annealing for 45 s at 65 °C for bacteria and 60 °C for archaea, and extension at 72 °C for 30 s, followed by a final extension period at 72 °C for 5 min. Sequencing was performed at the Göttingen Genomics Laboratory with a 454 GS-FLX sequencer and Titanium chemistry (Roche, Mannheim, Germany). Amplicon sequences were quality-filtered, denoised, clustered at 97% sequence, identity, chimera checked, and taxonomy was assigned using the SILVA database version 119[52] employing QIIME 1.8 scripts[53]. Singletons, chloroplasts, unclassified OTUs and extrinsic domain OTUs were removed by employing filter_otu_table.py. Rarefied OTU tables (bacteria 6800 and archaea 2000 sequences per sample) were used for the analyses.

**Net primary production**. The following components of net primary production (NPP) were measured from March 2013 to April 2014 on all 32 study plots: aboveground litterfall including pruned oil palm fronds, fine root production, rubber latex harvest, and oil palm fruit harvest, as well as stem increment. Litterfall from 16 litter traps on each plot was collected at monthly intervals and separated into leaves, woody material, propagules, and inflorescences, which were subsequently oven-dried for 72 h at 60 °C and weighted. In oil palm plantations all pruned palm fronds were counted and total dry weight extrapolated based on a dried subsample of fronds. To calculate woody biomass production based on the respective allometric equations[22] differences in tree aboveground biomass between census points were used. Manual dendrometer tapes (UMS, Munich, Germany) were mounted on 40 tree individuals per plot (960 in total) to obtain stem increment data. Oil palm biomass production was obtained from height increment data measured every 3 months. To estimate fine root production 16 ingrowth cores per plot were installed. After removal of the cores, root samples were processed in the same way as the root inventory samples.

**Organic carbon in plant biomass**. Stand structural parameters (height, diameter) were recorded for each tree with a diameter at breast height ≥ 10 cm on all 32 study plots using a Vertex III height meter (Haglöf, Långsele, Sweden). Wood density values were obtained from wood cores extracted from 204 trees. For the remaining trees, interpolated values derived from measurements of wood hardness with a Pilodyn 6 J (PROCEQ SA, Zürich, Switzerland) were applied. Allometric equations

were used to estimate aboveground woody biomass and coarse roots biomass for forest trees, rubber trees, and oil palms[22]. Fine roots biomass (diameter: ≤2 mm) was measured using 10 soil cores down to 50 cm soil depth at each plot. All fine roots segments > 1 cm length were extracted by washing on a sieve and separated under a stereomicroscope into live (biomass) and dead (necromass) fractions, pooled for the current analysis. The C content of each component (stem wood, fine roots, dead wood, rubber latex, oil palm fruit, all litter fractions) was analyzed with a CN auto-analyzer (Vario EL III, Hanau, Germany) and used to convert biomass into carbon units[22].

**Soil organic carbon and soil fertility indicators**. In 2013, soil samples were collected at three depth intervals (0.0–0.1 m, 0.1–0.3 m, 0.3–0.5 m) in each of the five randomly selected subplots per plot, and further three depth intervals (0.5–1.0 m, 1.0–1.5 m, and 1.5–2.0 m) at two of the five subplots. The mean of the five or two subplots represented the value for each replicate plot. Soil organic carbon for the 0.0–0.5 m and 0.5–2.0 m depths were cumulative stocks of the three depth intervals. For the soil fertility indicators (net N mineralization rate, extractable P, and exchangeable Ca, K, Mg, and Na), we used the measurements in the top 0.10 m depth. Soil organic C concentrations were analyzed from air-dried, ground soils using a CN analyzer (Vario EL Cube, Elementar Analysis Systems GmbH, Hanau, Germany). Net N mineralization was measured using an in situ buried bag method of intact soil cores. Extractable P was determined from air-dried, 2 mm sieved soils using the Bray 2 method. Exchangeable cations were determined by percolating air-dried, 2 mm sieved soils with unbuffered 1 M $NH_4Cl$ and cations were measured in percolates using an inductively coupled plasma-atomic emission spectrometer (ICP-AES; iCAP 6300 Duo VIEW ICP Spectrometer, Thermo Fischer Scientific GmbH, Dreieich, Germany).

**Soil respiration and soil greenhouse gas fluxes**. Soil $CO_2$, $CH_4$, and $N_2O$ fluxes were measured monthly for 1 year (2013) using vented, static chambers with permanently installed bases in four subplots per plot[20,54]. The mean of the four subplots represented the value for each replicate plot on each sampling period. During gas sampling, the chamber bases were closed and four gas samples (23 mL each) were taken at 1 min, 11 min, 21 min, and 31 min after chamber closure. Gas samples were immediately injected into pre-evacuated 12 mL Labco Exetainers and were analyzed using a gas chromatograph with electron capture and flame ionization detector (GC 6000 Vega Series 2, Carlo Erba Instruments, Milan, Italy). Soil gas fluxes were calculated from the linear increase of concentration over time of chamber closure and adjusted for the measured air temperature and pressure at the time of sampling.

**Nutrient-leaching fluxes**. Nutrient leaching was measured biweekly to monthly for 1 year (2013) using suction cup lysimeters (P80 ceramic, maximum pore size 1 μm; CeramTec AG, Marktredwitz, Germany), which were installed in two subplots per plot. These lysimeters were inserted into the soil down to 1.5 m depth. Soil water was withdrawn by applying a 40 kPa vacuum on the sampling tube. The collected soil water samples were stored in 100 mL plastic bottles and immediately frozen upon arrival at the field laboratory. Frozen water samples were transported to Germany and were kept frozen until analysis. The total dissolved N (TDN), $NH_4^+$, and $NO_3^-$ were measured using continuous flow injection colorimetry (SEAL Analytical AA3, SEAL Analytical GmbH, Norderstedt, Germany), whereas dissolved organic C was determined using a total organic carbon analyzer (TOC-Vwp, Shimadzu Europa GmbH, Duisburg, Germany). Dissolved Na, Ca, Mg, total Al, total P, and total S were analyzed using ICP-AES. Drainage water fluxes were estimated using a soil water model, parameterized with our measured site characteristics (climate data, leaf area index, rooting depth, soil water retention curve, texture, and bulk density)[23]. Element concentrations from each of the two lysimeters per replicate plot were multiplied with the total biweekly or monthly drainage water flux to get the nutrient-leaching fluxes. The annual leaching flux was calculated as the sum of biweekly to monthly measured leaching fluxes, and the average of the two lysimeters per plot represents the value of each replicate plot.

**Decomposition**. Litterbags with 10 g dry leaf litter mixture of three tree species were placed in each of the four land-use systems with one litterbag in each of the 32 study plots in October 2013 and retrieved in March 2014[55]. Litter mass loss was calculated as the difference between the initial dry mass and litter dry mass remaining after 6 months.

**Plant transpiration**. Plant transpiration was assessed at all 32 study sites by sap flux measurements with Granier-type thermal dissipation probes. The measurements were performed between March 2013 and March 2014 and lasted at least 3 weeks per site (52 days on average). Sap flux sampling and scaling schemes to stand transpiration (mm d$^{-1}$) differed for the four land-use types and were specifically adapted for forest and jungle rubber, rubber plantations and oil palm plantations[56,57]. The scaling scheme for forest, jungle rubber, and rubber included the application of radial sap flux profile functions with increasing stem xylem depth as derived from measurements with heat-field deformation sensors. Annual series of reference potential evapotranspiration calculated from micrometeorological measurements in the study region with the Priestly–Taylor equation were

subsequently used to extrapolate the transpiration series from each study site to the annual scale via a linear regression approach[57,58].

**Climatic conditions**. Microclimatic conditions were assessed with below canopy meteorological stations in each of the plots. They consisted of a thermohygrometer (Galltec Mella, Bondorf, Germany) placed at a height of 2 m above the ground to measure air temperature and air relative humidity and a soil sensor (IMKO Trime-PICO, Ettlingen, Germany) at a depth of 0.3 m to monitor soil temperature and soil volumetric moisture. Data were recorded hourly with a data logger (LogTrans16-GPRS, UIT, Dresden, Germany). Data covered the period June 2013 to October 2014.

**Farm household surveys (2012 and 2015)**. For the estimation of economic returns from land (profit), we analyzed data from a farm household survey conducted in five regencies in the lowlands of Jambi Province of Indonesia. The survey was carried out in two rounds; the first round in 2012 and the second in 2015[18]. For household selection, we used a multistage random sampling procedure. Five regencies (Sarolangun, Bungo, Tebo, Batanghari, and Muaro Jambi), which comprise most of the lowland transformation systems in Jambi, were selected purposively. From each of these regencies, we randomly selected four districts per regency and two rural villages per district, resulting in 40 randomly selected villages. In addition, five villages near to the Bukit Duabelas National Park and the Harapan Rainforest, where the ecological research was carried out, were purposively selected. Finally, we randomly selected farm households in the villages, based on household census data. In each village, we selected between 12 to 24 households, with the number adjusted to the total number of households residing in a village. In total, 701 households were interviewed in each round. For the 2015 round, we targeted the same households. Attrition rate was only at 6%. More than 50% of the sample households were from the regencies where ecological studies were conducted (Sarolangun and Batanghari). A structured questionnaire was administered to household heads through face-to-face interviews. Interviews were conducted by local enumerators who were trained and supervised by the researchers. Data on historical land-use changes and land transactions were collected alongside current management practices (e.g., inputs, output, market price, etc.). For profitability analysis, input-out data at the plot-level from the second round of household survey in 2015 were employed. We calculated annual profits per hectare by deducting production costs from plot revenues. Plot revenues were calculated by multiplying agricultural output with the average output prices per year received by farmers. Production costs were calculated by valuing all production factors and inputs by their usual market prices. While for wage labor and material inputs individual price data were collected, family labor was valued using the average agricultural wages in 2015. Wage levels were derived from the Indonesian Labor Survey (SAKERNAS). We calculated profits for jungle rubber, rubber and oil palm plots. For forest we set profits to zero, since only ~1% of the households marketed forest products in 2015. Annual profits for 2012 and 2015 (adjusted for inflation) are reported in Supplementary Table 2, illustrating the strong decrease in rubber prices from 2012 to 2015. We converted all profits from Indonesian rupiahs (IDR) to US dollars (USD), using the average exchange rate of the two currencies in 2015 (1.00 USD = 13389.413 IDR)[59].

**Qualitative interviews at household, sub-national, and national level**. To investigate the history of landscape transformation, spatial planning and the underlying political drivers of transformation we conducted semi-structured and open qualitative stakeholder interviews, focus group discussions and participatory rural appraisals at multiple levels. Research was inspired by multi-sited ethnography[60]. We followed the networks of different actors impacted by land-use transformation and land tenure conflicts and those driving transformation in Jambi. Interviews were conducted at the village and household scale including indigenous leaders and village governments in order to understand village and land-use history, impacts of state policies, individual land-use decisions, environmental change, and land tenure conflicts. At the sub-national and national level, we conducted interviews with state agencies, ministries, environmental NGOs and peasant and indigenous rights organizations. Qualitative research took place between 2012 and 2016. In total, we conducted 150 qualitative interviews.

**Estimation of profits from crop yields on the study plots and statistical analysis of biodiversity-profit and ecosystem function–profit relationships**. Linking biodiversity or ecosystem functions to the crop income required detailed information on farmers' profits at the plot level. However, because output prices and input prices vary significantly both spatially and temporally, restricting our analysis to the profits of the 24 plots on which the ecological studies were conducted (including yield measurements of rubber and oil palm) would likely generate biased estimates. Instead, we followed a two-step procedure to estimate profits at level of the ecological study plots. First, we established the relationships between crop yields and profits for the three land-use systems jungle rubber, rubber, and oil palm as based on the household surveys. Second, we predicted the profits from yields on the ecological study plots from these relationships, including the variability in the relationships between yields and profits established in the first step. In the following, we describe our approach in more detail. Using the included

information on yields (annual harvest amounts) and profits per ha of smallholder farmers, we modeled yield–profit relationships with simple linear regression. We found evidence of strong, positive, and linear relationships between the yields and farmer's profits for all the three crops (Supplementary Fig. 4). Likewise, increasing variances in profits for increasing yields were consistently observed irrespective of the crop. Second, we used the model coefficients of these relationships to predict the profits of the 24 jungle rubber, rubber, or oil palm plots based on the yields at plot level (see section "Yield assessment"). We did not restrict ourselves to predictions of their average profits but also included estimations of the heteroscedastic residual variances in the yield–profit relationship of each crop (i.e., non-constant variances of the deviations of observed data points from model predictions; Supplementary Fig. 4). Note, however, that neither the predicted average profits nor the average profits plus a randomly drawn error term correspond to the true, unobserved profits. Hence, profits can be seen as a variable exhibiting measurement error. It is known that measurement error in explanatory variables leads to downward biased regression coefficient and predictions.

To this end, we applied the simulation and extrapolation (SIMEX) method[61]. The idea of the SIMEX method is to exploit the relationship between different degrees of measurement error variances and the bias of the estimators of interest. Let $\sigma_e^2$ be the measurement error variance in the explanatory variable and $\beta$ a single parameter of interest associated with this variable. Let further $G(\sigma_e^2)$ be a function describing the relationship between the potentially biased estimator of $\beta$ for infinitely large sample size and $\sigma_e^2$. For different values of $\lambda, \lambda \geq 0$, additional measurement error with variance $\lambda\sigma_e^2$ is added to the explanatory variable with measurement error, resulting in a measurement error variance of $(1 + \lambda)\sigma_e^2$. Given a certain value for $\lambda$, the measurement error is simulated $B$ times and the parameter of interest is estimated in each of the $B$ steps while not accounting for the measurement error. Averaging over the $B$ estimates yields convergence to $G$. Applying this bootstrap procedure for different values of $\lambda$, the relationship between the degree of measurement error and the resulting estimator $G$ of interest can be estimated, e.g., in a linear or quadratic fashion using ordinary least squares. Eventually, the predicted value of this function for no measurement error, i.e., $\lambda = -1$, is calculated, which is the SIMEX estimator $\beta_{SIMEX} = G(0)$.

In our application, the heteroscedastic measurement error variances for the three crops were estimated from the yield–profit relationships as described above. We used a grid of 10 equidistant values for $\lambda$ between 0.1 and 3, $B = 200$ bootstrap replication for each $\lambda$ and a quadratic fit to model the relationship between different degrees of measurement error variances and the bias of the estimators of interest. We were interested in the coefficients from the links between profit and biodiversity or ecosystem functions, which were modeled in a non-linear fashion via penalized splines within the generalized additive models framework. The negative binomial distribution was chosen to account for the count data nature with potential overdispersion in the case of models with species richness as response. We used the R package *simex* (ver. 1.7)[62] that includes the implementation of the SIMEX method for generalized additive models with heteroscedastic measurement error in the explanatory variable. In summary, this approach allowed us to capture the measurement error in the profits and to unbiasedly estimate the relationships between profit and the different biodiversity or ecosystem functions.

We modeled the relationships between biodiversity and profit for each studied taxonomic group, using two measures of species richness as response, respectively: (1) richness based on all species recorded in a study plot and (2) richness based only on those species that also were recorded in forest plots. The relationships between ecosystem functions and profit were separately modeled for each indicator variable per function.

### Calculation of multidiversity and multifunctionality and their relationships to profits.

We calculated indices of multidiversity and multifunctionality in order to test if the observed trade-offs between economic and ecological indicators at the level of individual taxonomic groups and ecosystem functions are also evident when considering all groups or functions simultaneously (see Supplementary Table 3 for an example calculation). Multidiversity was calculated based on species richness of all 14 studied taxonomic groups; likewise, we calculated multifunctionality based on all 36 indicators of the 10 studied ecosystem functions. These indices are commonly calculated as the proportion of plot-level measured functions or taxonomic groups of which performance exceeds an a priori minimum defined threshold (e.g., high, medium, or low performance) as compared with the maximum measured performance level[14]. The maximum performance level is thereby not restricted to a specific land-use system, i.e., although some functions may peak in forest plots, others may peak in monoculture plantations. Since defining a specific threshold that determines whether a given ecosystem function or taxonomic group contributes to multifunctionality or multidiversity can be arbitrary, we calculated the full range of thresholds from 1% to 99%[14,26]. This approach also allowed us to investigate whether relationships with profitability differed depending on expectation levels of minimum ecosystem functioning or biodiversity performance, which is particularly relevant for defining goals of landscape management; for example, management expectations of multifunctionality based on a 90% threshold are much more stringent than multifunctionality based on a 50% threshold. First, we defined the 100%-level of biodiversity (species richness per taxonomic group) or ecosystem functioning for each group or function as the mean

of the five highest recorded values to reduce potential influence of outliers[63]. Second, a threshold was defined at which levels of species richness or ecosystem function performance were considered sufficiently high to contribute to local multidiversity or multifunctionality, respectively. For example, when considering a threshold of 50%, only those species groups that are at least at 50% of their average maximum observed species richness (across all plots) will contribute to local multidiversity. Multidiversity and multifunctionality were then defined as the proportion of species groups or functions that locally exceeded the threshold as compared with the total number of species groups or functions that were studied in a given study plot (see Supplementary Table 3 for an example calculation of multifunctionality based on three functions). Because for most ecosystem functions multiple indicators were measured, we weighted these indicators according to their proportional share on the function (e.g., for an ecosystem function measured with eight indicators, each indicator variable was weighted 12.5%). Furthermore, we used the inverse of the indicators for which high values indicated less desirable functioning (e.g., nutrient leaching and soil greenhouse gas fluxes; see Fig. 3), so that in all cases, high values indicated high levels of ecosystem functions contributing to multifunctionality. We calculated multidiversity and multifunctionality for all study plots and across the full range of thresholds from 1% to 99% at steps of 0.01%. We then related the index values to the average predicted profits of crop yields at plot-level with simple linear regression.

### Landscape composition optimization using a genetic algorithm.

To identify how landscape design may be optimized to mitigate the observed socioeconomic–ecological trade-offs at landscape scales, we designed conceptual in silico landscapes of different composition, using the four studied land-use systems (forest, jungle rubber, rubber monoculture, oil palm monoculture) as input. These "optimized landscapes" were informed by the plot-level data from the ecological surveys and associated profit estimates. A virtual landscape consisted of 32 empty 50 m × 50 m slots, corresponding to the number of empirical study plots and their spatial extent in our ecological surveys. For four taxonomic groups, data were only available for a reduced number of study plots (archaea and fungi: 30 plots; testate amoebae and birds: 31 plots); for these cases, the landscape size was adjusted accordingly. We then "filled" the empty slots to identify the landscape composition that resulted in the highest-possible level of multidiversity, multifunctionality, species richness (total richness across all included plots), or ecosystem functioning (sum of standardized and equally-weighted indicator values of an ecosystem function) at the scale of the conceptual landscape. Filling was done using the plot-level data with replacement, i.e., the solution space for the composition of the virtual landscape encompassed all possible combinations from landscapes that were made up by repeated fills with only one study plot to landscapes consisting of combinations of all 32 plots. To incorporate potential socioeconomic–ecological trade-offs into this optimization process, we constraint the landscape solutions by a priori defined minimum profit expectations. In line with the expectation that farmers and landscape managers aim at increasing profits from land-use, we optimized the landscapes for six profit expectations, whereby the expected profit was the average profit of the included plots: 0, 200, 400, 600, 800, 1000 USD ha$^{-1}$ year$^{-1}$. These expectations corresponded to the average profits estimated from our yield assessments (Supplementary Fig. 4). By constraining the optimization process to these expectations, a landscape solution was allowed to surpass a given expectation, but solutions that did not provide the expected profit (e.g., because a conceptual landscape consisted predominantly of forest plots) were discarded.

Because the number of solutions of a landscape with 32 plots that are filled with replacement is very large ($\sim 9.16 \times 10^{17}$), a brute force approach whereby all possibly solutions are calculated to identify the best solution (i.e., the landscape composition that most efficiently minimizes the trade-off, or the Pareto-frontier) was not computationally feasible. Instead, we used a binary genetic algorithm (GA) for the optimization process. Genetic algorithms mimic evolutionary processes to solve otherwise numerically or computationally non-solvable discrete and continuous optimization problems[64]. They have been suggested as heuristic optimization techniques to support landscape design for optimal profit-natural value relationships[65,66]. As in natural evolution, GAs code information within genes (located on chromosomes), individuals (bearing chromosomes) and populations of individuals. Exploration and exploitation facilitate the evolutionary process of the GA[64]. Exploration of the parameter space is achieved by mutation of genes and cross-over of genetic information between chromosomes of individuals of a founder population. Exploitation mimics the "survival of the fittest" observed in natural populations, and reduces the diversity in the population by selecting the fittest individuals for the next generation while discarding poor performers to make room for new offspring. The resulting optimization process makes GAs powerful tools to solve large and complex computational problems. In our binary GA, the decision whether to include or not to include a study plot was coded as a gene with 1 = inclusion and 0 = no inclusion. Because each landscape consisted of 32 study plots (or 30–31, see above) and our approach allowed for filling with replacement, each gene needed to be replicated 32 times. The resulting 1024 genes were arranged on one chromosome. Each chromosome represented a landscape solution with defined landscape composition (coded by the 0 s and 1 s that inform whether a plot is included or not) and consequently defined profit, biodiversity and ecosystem functioning.

We used the above-described GA to identify optimized landscape compositions for each target variable, i.e., species richness of taxonomic groups, performance of ecosystem functions as weighted averages of function indicators, multidiversity, and multifunctionality. In each GA optimization, the population size was 500 chromosomes and the optimization process continued for 100 generations before selecting the best landscape solution.

We used the package "genalg" version 0.2.0[67] to implement the GAs in the R statistical environment[68].

**Reporting summary**. Further information on research design is available in the Nature Research Reporting Summary linked to this article.

## Data availability

The data that support the findings of this study are available from github, https://github.com/ingograss/sumatra_landuse_tradeoffs. The source data underlying Figs. 2–5 are provided as a Source Data file.

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

## Acknowledgements

We thank the village leaders, respondents, farmers, PT REKI and Bukit Duabelas National Park for granting us access to sites and information, and Ristekdikti for the research permits. Rizky Nazarreta, Herry Marta Saputra and Azru Azhar (all IPB Bogor) assisted with identification of canopy ants, Anik Larasati, Cici Indriani, Nurul Novianti, Iqbal Tawakkal (all IPB Bogor) with identification of canopy parasitoid wasps, Peggie Djunijanti (LIPI Cibinong) with the identification of butterflies. We thank Christoph Kleinn for comments on a previous version of this manuscript. This study was financed by the Deutsche Forschungsgemeinschaft (DFG), project number 192626868, in the framework of the collaborative German–Indonesian research center CRC990.

## Author contributions

I.G. led the writing of the paper. I.G. and P.P. conducted the analysis. I.G., C.K., V.V.K, M.D.C., O.M., P.P., J.D., K.R., and S.S. designed the study and assembled the data sets. I.G., C.K., V.V.K, M.D.C., O.M., P.P., J.D., K.R., E.S.A., A.D.B., N.B., U.B., B.B., D.B., R.D., K.F.A.D., H.F., L.F., J.H., N.H., P.H., D.H., M.J., A.K., M.M.K., V.K., H.K., C.L., N.J.S.L., R.P., A.P., A.M.P., E.P., M.Q., A.R., S.S., D.S., A.T., T.T., E.V., and M.W. contributed data and participated in discussions on study design, analysis, and interpretation. All authors contributed to writing.

## Competing interests

The authors declare no competing interests.

## Additional information

Ingo Grass [1,2]✉, Christoph Kubitza[3,4], Vijesh V. Krishna [5], Marife D. Corre[6,7], Oliver Mußhoff[3,7], Peter Pütz[8], Jochen Drescher [9], Katja Rembold[10,11], Eka Sulpin Ariyanti[12], Andrew D. Barnes [13], Nicole Brinkmann[14], Ulrich Brose [15,16], Bernhard Brümmer[3,7], Damayanti Buchori[17], Rolf Daniel [18], Kevin F.A. Darras[2], Heiko Faust [7,19], Lutz Fehrmann[20], Jonas Hein [21], Nina Hennings[22], Purnama Hidayat[23], Dirk Hölscher[7,24], Malte Jochum [9,25,26], Alexander Knohl [7,27], Martyna M. Kotowska [28], Valentyna Krashevska[9], Holger Kreft[7,10], Christoph Leuschner[7,28], Neil Jun S. Lobite[29], Rawati Panjaitan[23], Andrea Polle [7,14], Anton M. Potapov[9,30], Edwine Purnama[20], Matin Qaim [3,7], Alexander Röll [24], Stefan Scheu[7,9], Dominik Schneider[18], Aiyen Tjoa[31], Teja Tscharntke[2,7], Edzo Veldkamp [6,7] & Meike Wollni[3,7]

[1]Ecology of Tropical Agricultural Systems, University of Hohenheim, Garbenstrasse 13, 70599 Stuttgart, Germany. [2]Agroecology, University of Göttingen, Grisebachstrasse 6, 37077 Göttingen, Germany. [3]Department of Agricultural Economics and Rural Development, University of Göttingen, Platz der Göttinger Sieben 5, 37073 Göttingen, Germany. [4]German Institute of Global and Area Studies (GIGA), Neuer Jungfernstieg 21, 20354

Hamburg, Germany. [5]International Maize and Wheat Improvement Center (CIMMYT), Carretera México-Veracruz Km. 45, El Batán, Mexico. [6]Soil Science of Tropical and Subtropical Ecosystems, University of Göttingen, Büsgenweg 2, 37077 Göttingen, Germany. [7]Centre of Biodiversity and Sustainable Land Use (CBL), University of Göttingen, Büsgenweg 1, 37077 Göttingen, Germany. [8]Chair of Statistics, Faculty of Economic Sciences, University of Göttingen, Humboldtallee 3, 37073 Göttingen, Germany. [9]Department of Animal Ecology, J.F. Blumenbach Institute of Zoology and Anthropology, University of Göttingen, Untere Karspüle 2, 37073 Göttingen, Germany. [10]Biodiversity, Macroecology & Biogeography, University of Göttingen, Büsgenweg 1, 37077 Göttingen, Germany. [11]Botanical Garden of the University of Bern, Altenbergrain 21, 3013 Bern, Switzerland. [12]Magister of Environmental of Science, University of Lampung, Lampung 35145, Indonesia. [13]School of Science, University of Waikato, Private Bag 3105, Hamilton 3240, New Zealand. [14]Forest Botany and Tree Physiology, University of Göttingen, Büsgenweg 2, 37077 Göttingen, Germany. [15]EcoNetLab, German Centre for Integrative Biodiversity Research (iDiv) Halle-Jena-Leipzig, Deutscher Platz 5e, 04103 Leipzig, Germany. [16]EcoNetLab, Friedrich Schiller University Jena, Dornburger-Str. 159, 07743 Jena, Germany. [17]Center for Transdisciplinary and Sustainability Sciences, IPB University, Bogor Agricultural University, Jalan Pajajaran, Bogor 16128, Indonesia. [18]Department of Genomic and Applied Microbiology and Göttingen Genomics Laboratory, University of Göttingen, Grisebachstr. 8, 37077 Göttingen, Germany. [19]Human Geography, University of Göttingen, Goldschmidtstr. 5, Göttingen, Germany. [20]Forest Inventory and Remote Sensing, University of Göttingen, Büsgenweg 5, 37077 Göttingen, Germany. [21]Institute of Geography, Kiel University, Ludewig-Meyn-Str. 14, 24118 Kiel, Germany. [22]Soil Science of Temperate Ecosystems, University of Göttingen, Büsgenweg 2, 37077 Göttingen, Germany. [23]Department of Plant Protection, Faculty of Agriculture, Bogor Agriculture University, Jln. Kamper, Kampus IPB Dramaga, Bogor 16880, Indonesia. [24]Tropical Silviculture and Forest Ecology, University of Göttingen, Büsgenweg 1, 37077 Göttingen, Germany. [25]Experimental Interaction Ecology, German Centre for Integrative Biodiversity Research (iDiv) Halle-Jena-Leipzig, Deutscher Platz 5e, 04103 Leipzig, Germany. [26]Institute of Biology, Leipzig University, Deutscher Platz 5e, 04103 Leipzig, Germany. [27]Bioclimatology, University of Göttingen, Büsgenweg 2, 37077 Göttingen, Germany. [28]Plant Ecology and Ecosystems Research, University of Göttingen, Untere Karspüle 2, 37073 Göttingen, Germany. [29]Animal Biology Division, Institute of Biological Science, University of the Philippines, Los Baños 4031, Philippines. [30]A.N. Severtsov Institute of Ecology and Evolution, Russian Academy of Sciences, Leninsky Prospect 33, 119071 Moscow, Russia. [31]Agriculture Faculty, Tadulako University, Jl. Soekarno Hatta km.09, Tondo, Palu, Indonesia. ✉email: ingo.grass@uni-hohenheim.de

