## [Peer Review File · Nature Communications]

Reviewers' comments:

Reviewer #1 (Remarks to the Author):

This paper is an impressive collection of studies designed to answer how small-scale agricultural plots transitioning from diversified agroforestry systems to monocultures impacts biodiversity and ecosystem function. The study demonstrates generally negative relationships between the profits gained from monocultures and several biodiversity and ecosystem function metrics. The paper is written well, the results are novel and presented clearly. The statistical analyses and modeling approaches seem solid. I only have a few minor comments.

1. Though the term multidiversity is starting to make inroads, I think it is fairly uncommon and should be explained better. It is just biodiversity across several taxonomic groups as far as I can tell, perhaps a citation and a bit clearer text here.
2. There is some surprising spread at the high end in the profit-yield graph of Ex. Fig 5. I think this warrants some suggestion esp. considering the strength of the statement in Line 68. What is happening to the very high yielding farms making such little money?

Line 4: Ecosystem multidiversity? Do you mean multifunctionality here?

Line 82/3: combination... is a bit awkward followed by three terms.

Line 93: Multidiversity is a phrase I have not come across often, the definition here, whole system biodiversity, is confusing. Authors should provide a citation and explain what the whole system is.

Line 130: Confusing – or even positively-, rephrase?

Line 153: Some case studies for support would be useful, citations?

Extended Data Fig. 1-3: No explanation of red lines in these figures. Should be repeated for each.

Extended Fig 5: Spread on data is surprising. It would be interesting to discuss why some high yield farms are so unprofitable.

Reviewer #2 (Remarks to the Author):

This paper aims to quantify the economic-ecological trade-offs associated with land use transitions from forest to tree crops, namely rubber and oil palm. This is a valuable area of research, nationally, regionally, and globally, given (a) the acute and widespread negative environment consequences of extensive agricultural expansion, (b) the importance of these crops for global poverty and economic development in the global south, where they are predominantly produced, and (c) the importance of these crops for the global food system and food security. While these three issues have been well documented independently, to my knowledge this is the first—if not the first, the most rigorous and most convincing—study to links biodiversity and multi-functionality in these landscapes to the underlying economic incentives, thus marking a major contribution to the literature and to public policy.

The authors bring together an innovative and extremely impressive combination of data spanning that used in environmental science and qualitative and quantitative social sciences. Though I am not an ecologist, the statistical methods used appear sound and, honestly, after parsing many studies on these topics at the nexus of environmental and social sciences I have not seen an exercise like this (e.g., on palm oil trade-offs) undertaken with this level of rigour. The supporting files and figures are clean and the paper's results are, for the most part, presented in a clear and convincing way. I enjoyed reading the paper, and only have one overall comment and a few specific comments. I hope they are helpful.

Overall comment

The economic impacts reach far beyond those in the profit function assessed here, as is indeed evidence in some of the authors' other studies (the same goes for the environmental impacts). Thus the interpretation of overall trade-offs remains defined by the scope of the items considered on each side of the ledger. I believe this should be stressed in the paper more: that any exercise of this nature will always be only as complete as what is included, and as the scope of things included increases, the precision likely decreases, and the uncertainty with which we can make overall assessments increases quite a lot. I am also concerned about the profit functions, specifically, that we can't generalize from the conditions on Jambi across Indonesia or to industrial estate systems, and that there are probably still simultaneity and omitted variable concerns affecting parameter bias here, even after addressing measurement error. However, I think the SIMEX method is an appropriate response and note that is but one small part of a much large constellation of quantitative analyses in the paper.

Specific comments

P2 L64 As a non-ecologist, defining "multi-diversity" would have been helpful for me.

P2 L66 Can the genetic algorithm be described in plain English? If not, you might consider removing it from the high-level summary.

Pg 2 L 69 “can only be reduced” seems too strong. Strong governance and regulation around spatial planning is one way that public policy can curb biodiversity loss and land use change instead of appealing to set the price right for the market to do the work.

Pg 4 L 186 You might briefly explain what pareto-optimum means for non-economists.

Pg 4 L 196 How do these findings reconcile with other work by the authors that suggests palm oil is more profitable than rubber, and of course the revealed preference of those shifting?

Pg 6 L 244 I think this study is incredibly valuable but it does not tell us what specific policy interventions will work best to manage the trade-offs. The authors have three options here: mention specific interventions, mention vague policy directions, or say nothing. Vague directions are weaker than being specific but you do not have the evidence in this paper to support being specific. I do not think you should be using studies from other contexts to justify advocating particular interventions, as we frankly still know little about what works and cannot generalize these past studies over. What this study does is help to highlight the primacy of economic drivers here and the necessity of policy design accounting for these (for example, though PES type approaches) or neutralizing them (through top-down regulations). If I were the authors, I would stop short of recommending (c.f., floating as options) any particular solution as you have not trialled any such things in the paper. Thus, this last discussion could perhaps be pared back. It is certainly the weakest part of the paper, in my view, even though I broadly agree with the positions staked and directions proposed.

Ryan Edwards

Reviewer #3 (Remarks to the Author):

The paper tackles a crucial question about trade-offs in biodiversity and livelihood profits for smallholders, and the best designs for mitigating trade-offs in these landscapes. The paper builds on other published studies by the research team at their Sumatra site analysing the same/similar data sets (e.g. Clough et al. Nature Comms 2016).

The focus of the paper on ecological-economic trade-offs is very important. The authors present a large amount of information from a smallholder landscape of 700 farms cultivating rubber or oil palm. They conclude that biodiversity conservation requires economic incentives to be changed in order to protect remaining areas of forest. This is an important finding, but not a new conclusion.

The focus on smallholders is important because these landscapes are generally less well-studied than industrial plantations – and these smallholder producers comprise a large proportion of agricultural landscapes and make large contributions to global rubber and oil palm production, particularly on Sumatra. The paper presents a substantial piece of work, reporting data from 14 taxa and 10 ecological functions from 700 smallholder farms.

The trade-offs the authors report occur because essentially two of the habitat types they study have high biodiversity/desirable function but no/low profit (degraded forest and jungle rubber), whereas the other two habitats are monocultures of rubber and oil palm which have low biodiversity/function but high profit. The genetic algorithm modelling confirms that selecting landscapes to maintain high multidiversity results in more forest, whilst maintaining high profit results in more oil palm. The solution for multifunctionality are less straightforward, with no consensus solution to maintain all functions. The authors then go on to suggest PES/REDD in order to provide enhanced livelihood benefits from forest.

This paper includes a huge amount of field data, carefully presented to address an important problem. The paper builds on other published studies from the team/study system. The authors should be congratulated on their field work efforts and synthesis. However, I am unclear what is conceptually novel in their study – we know from many studies (including the authors' works) that tropical forests are hyper diverse, such that land-use change for agriculture (esp. rubber and oil palm etc.) leads to declines. Hence the study arguably adds to the current knowledge on land-use change impacts and ecological-economic trade-offs, but provides no new understanding. It would have been helpful if the authors could have explained more clearly at the outset about the novelty of their work, and what new conclusions now arise compared with theirs and others' previous studies. I am not sure what the modelling of theoretical landscapes adds to the understanding of landscape change beyond that presented in the empirical data analyses (i.e. less forest = more profit but less diversity). I think this novelty should be much more clearly stated. The final discussion section around REDD and PES solutions also does not move our understanding forward, and REDD/PES arguments are known to rely on very high prices for carbon to match economic benefits from agriculture, much higher prices than current carbon markets. This point has been raised in many other studies.

So I think this is a very valuable data set, but suggest the authors take more care to explain what is already known (including in their study system), what is not, and how their study address new topics e.g. L80 – simply stating its 'poorly studied' is a weak justification for the study aims. Major restructuring of the paper to focus on novel investigations and findings would produce a much stronger and valuable paper.

Reviewers' comments:

Reviewer #1 (Remarks to the Author):

This paper is an impressive collection of studies designed to answer how small-scale agricultural plots transitioning from diversified agroforestry systems to monocultures impacts biodiversity and ecosystem function. The study demonstrates generally negative relationships between the profits gained from monocultures and several biodiversity and ecosystem function metrics. The paper is written well, the results are novel and presented clearly. The statistical analyses and modeling approaches seem solid. I only have a few minor comments.

RESPONSE: Thank you very much for your helpful comments and appreciation of our work. Please see our response to each of your comments below.

1. Though the term multidiversity is starting to make inroads, I think it is fairly uncommon and should be explained better. It is just biodiversity across several taxonomic groups as far as I can tell, perhaps a citation and a bit clearer text here.

RESPONSE: We apologize for the confusion. What we mean is whole-ECOsysteM biodiversity of land-use types, i.e. the total diversity of species in a given land-use type (forest, jungle rubber, rubber, oil palm). We changed the wording accordingly. We also added references for multidiversity (Allan et al. 2014 PNAS) and multifunctionality (Byrnes et al. 2014 Methods in Ecology and Evolution). A more detailed definition of both multidiversity and multifunctionality is provided in L168-181 and their calculations are furthermore detailed in the Supplementary Information. In both cases we used the same mathematical framework, i.e. a threshold-based approach sensu Byrnes et al. 2014 Methods in Ecology and Evolution.

2. There is some surprising spread at the high end in the profit-yield graph of Ex. Fig 5. I think this warrants some suggestion esp. considering the strength of the statement in Line 68. What is happening to the very high yielding farms making such little money?

RESPONSE: This is indeed an interesting question. We discussed it in our consortium and also looked into the data on these specific cases. The explanations differ and are strongly case-dependent. In one case, very high labour costs (about 4x the labour employed when compared to other farms of similar size) had the effect that the farmer did not earn profits from the crop, despite the high yields that resulted from the substantial labour force. In other cases, farmers heavily invested into inputs (herbicides, pesticides). A few farmers also reported high profits compared to their yields as they were able to attain above average output prices. As some of these farmers reported also very high yields, the high output prices and high yields led to some cases with substantial profits. Although these cases add to the spread at the high end of the profit-yield relationship, they are few in number and therefore do not markedly affect the estimations of the profit-yield relationships. We added the following sentence to the caption of Extended Data. Fig. 4: "At very high yield levels, a few farmers had higher or lower than expected profits (especially for oil palm), which is due to above-average output prices they obtained or above-average input and labor costs".

Line 4: Ecosystem multidiversity? Do you mean multifunctionality here?

RESPONSE: This comment was probably directed at the phrase in L64 “for 26,894 aboveground and belowground species and ecosystem multidiversity“. Here, we refer to the multidiversity of the whole ecosystems, i.e. the total diversity of all aboveground and belowground species. We rephrased the sentence accordingly: “aboveground and belowground species and whole-ecosystem multidiversity”.

Line 82/3: combination... is a bit awkward followed by three terms.

RESPONSE: We shortened the sentence to two terms: “which hold the potential to combine high yields and high biodiversity“.

Line 93: Multidiversity is a phrase I have not come across often, the definition here, whole system biodiversity, is confusing. Authors should provide a citation and explain what the whole system is.

RESPONSE: Please see our response to this question above.

Line 130: Confusing – or even positively-, rephrase?

RESPONSE: Rephrased to “While the total species richness of a few groups was not related to or even increased with higher profits [...]“ (L145-146).

Line 153: Some case studies for support would be useful, citations?

RESPONSE: We agree and added citations to four case studies on environmental functions in our study system to illustrate their complex responses to land-use change:

- *Kotowska et al. 2015 Global Change Biology (above- and belowground carbon storage in biomass across forest, jungle rubber, rubber and oil palm)*
- *Röll et al. 2019 Agricultural and Forest Meteorology (plant transpiration across the four land-use systems)*
- *Meijide et al 2018 Agricultural and Forest Meteorology (impact of forest conversion to monocultures on microclimate)*
- *Kurniawan et al. 2018 Biogeosciences (nutrient leaching losses and nutrient retention efficiency in highly weathered soils)*

Extended Data Fig. 1-3: No explanation of red lines in these figures. Should be repeated for each.

RESPONSE: We extended the figure legends for all Extended Data Fig. 1-3.

Extended Fig 5: Spread on data is surprising. It would be interesting to discuss why some high yield farms are so unprofitable.

RESPONSE: Please see our response to this question above.

Reviewer #2 (Remarks to the Author):

This paper aims to quantify the economic-ecological trade-offs associated with land use transitions from forest to tree crops, namely rubber and oil palm. This is a valuable area of research, nationally, regionally, and globally, given (a) the acute and widespread negative environment consequences of extensive agricultural expansion, (b) the importance of these crops for global poverty and economic development in the global south, where they are predominantly produced, and (c) the importance of these crops for the global food system and food security. While these three issues have been well documented independently, to my knowledge this is the first—if not the first, the most rigorous and most convincing—study to link biodiversity and multi-functionality in these landscapes to the underlying economic incentives, thus marking a major contribution to the literature and to public policy.

The authors bring together an innovative and extremely impressive combination of data spanning that used in environmental science and qualitative and quantitative social sciences. Though I am not an ecologist, the statistical methods used appear sound and, honestly, after parsing many studies on these topics at the nexus of environmental and social sciences I have not seen an exercise like this (e.g., on palm oil trade-offs) undertaken with this level of rigour. The supporting files and figures are clean and the paper's results are, for the most part, presented in a clear and convincing way. I enjoyed reading the paper, and only have one overall comment and a few specific comments. I hope they are helpful.

RESPONSE: Thank you very much for these very positive remarks and your helpful comments. Please see our responses to each of them below.

The economic impacts reach far beyond those in the profit function assessed here, as is indeed evidence in some of the authors' other studies (the same goes for the environmental impacts). Thus the interpretation of overall trade-offs remains defined by the scope of the items considered on each side of the ledger. I believe this should be stressed in the paper more: that any exercise of this nature will always be only as complete as what is included, and as the scope of things included increases, the precision likely decreases, and the uncertainty with which we can make overall assessments increases quite a lot.

RESPONSE: Thank you highlighting this important point. It is of course true that the economic impacts reach far beyond those which we assess in the profit function in the current manuscript. Indeed, we have studied these additional dimensions in great details, including but not limited to household income, consumption expenditures, food security, and nutrition:

- *Kubitza, C., V.V. Krishna, Z. Alamsyah, M. Qaim (2018). The Economics behind an Ecological Crisis: Livelihood Effects of Oil Palm Expansion in Sumatra, Indonesia. Human Ecology, Vol. 46, pp. 107-116.*
- *Krishna, V.V., M. Euler, H. Siregar, M. Qaim (2017). Differential Livelihood Impacts of Oil Palm Expansion in Indonesia. Agricultural Economics, Vol. 48, No. 5, pp. 639-653.*

- Euler, M., V. Krishna, S. Schwarze, H. Siregar, M. Qaim (2017). *Oil Palm Adoption, Household Welfare, and Nutrition among Smallholder Farmers in Indonesia*. *World Development*, Vol. 93, No. 1, pp. 219-235.

However, please note that these other dimensions are measured at the household level, not the field level, and cannot reasonably be expressed per unit of land, which is important in this manuscript to quantify the trade-offs with ecological functions. This is why we focus on profit here, which can be expressed per unit of land. Fortunately – as our other papers show – profit per hectare is positively associated with other economic and human welfare dimensions (income, food security etc.), so that it is a suitable proxy for wider human welfare in this particular case. We included this very important point in the revised manuscript (Introduction + Discussion) and added the above-listed publications as additional references:

- L129-L132 in the Introduction: “We focused on profits as these can be expressed per unit of land, and profits are positively associated with other economic and human welfare dimensions in our study system, such as household incomes, food security and consumption expenditures of smallholders^{17,18}.”
- L254-259 in the Discussion: “The economic impacts of the oil palm boom for smallholders reach far beyond those which we assessed in the profit function of this study. Indeed, adoption of oil palm production has not only increased household incomes, but also enhanced food security, nutrition and consumption expenditures of adopting smallholder farmers in Jambi Province^{17,18,32}. At national level, it estimated that the oil palm boom since 2000 may lifted up to 2.6 million rural Indonesians out of poverty³³.”

We are also aware that the oil palm expansion may have contributed to the development of infrastructure through additional tax revenues. In addition, economic spillovers from agricultural processing (palm oil mills) might have further incentivized local economic development. However, these dimensions can again not be reasonably expressed per unit of land.

Edwards RB. (2019). Export agriculture and rural poverty: evidence from Indonesian palm oil. Working Paper, Dartmouth College, USA

Edwards RB. (2019). Spillovers from agricultural processing. Working Paper, Dartmouth College, USA

Kubitza C., Gehrke E. (2018). Why does a labor-saving technology decrease fertility rates? Evidence from the oil palm boom in Indonesia. EForTS Discussion Paper 22, University of Goettingen

I am also concerned about the profit functions, specifically, that we can't generalize from the conditions on Jambi across Indonesia or to industrial estate systems, and that there are probably still simultaneity and omitted variable concerns affecting parameter bias here, even after addressing measurement error. However, I think the SIMEX method is an appropriate response and note that is but one small part of a much large constellation of quantitative analyses in the paper.

RESPONSE: We agree that it is likely that the estimated yield-profit relationships may only hold for Jambi's smallholder farmers. However, as the reviewer also acknowledges, these relationships are estimated reliably and there is not an endogeneity problem (which can only occur if we want to identify causal effects) as we only use the parameter estimates for prediction purposes. On a more general note, while the socioeconomic data collected in Jambi are representative of the Province, the evidence from Jambi cannot directly be extrapolated to other regions and countries. Especially the effects of oil palm expansion on income distribution may differ depending on the local context. In Jambi, much of the rainforest had already been cleared and local farmers were used to the production of commercial cash crops before oil palm was introduced in the 1980s. In other regions, where local communities are more dependent on forests and subsistence agriculture, the distribution of benefits would likely be different (Santika et al. 2019). However, the more general findings from Jambi, namely that oil palm expansion has contributed to poverty reduction and economic welfare gains among farm and non-farm households, are consistent with recent studies that used nationally representative data from Indonesia (Edwards 2019, Kubitza et al. 2019, Kubitza & Gehrke 2018).

We added the following sentences in the concluding part of the manuscript (L252-254): “The concrete results reported here are specific for Jambi. However, while some of the details may differ by region, the general findings on the economic-ecological tradeoffs will likely also hold for other parts of Indonesia and tropical lowland regions worldwide.”

Edwards RB (2019). Export agriculture and rural poverty: evidence from Indonesian palm oil. Working Paper, Dartmouth College, USA

Kubitza C, Gehrke E (2018). Why does a labor-saving technology decrease fertility rates? Evidence from the oil palm boom in Indonesia. EFForTS Discussion Paper 22, University of Goettingen

Kubitza C, Bou Dib J, Kopp T, Krishna VV, Nuryanto N, et al. (2019). Labor savings in agriculture and inequality at different spatial scales: The expansion of oil palm in Indonesia. EFForTS Discussion Paper 26, University of Goettingen

Santika T, Wilson KA, Budiharta S, Law EA, Poh TM, et al. (2019). Does oil palm agriculture help alleviate poverty? A multidimensional counterfactual assessment of oil palm development in Indonesia. World Development 120:105–17

Specific comments

P2 L64 As a non-ecologist, defining “multi-diversity” would have been helpful for me.

RESPONSE: We added further explanation and references to the manuscript. What we mean with “multidiversity” is the whole-ecosystem biodiversity of land-use types sensu Allan et al. (2014), PNAS. Thus, multidiversity is the total diversity of species in a given land-use type (forest, jungle rubber, rubber, oil palm). We changed the wording accordingly. We also added references for the definition and terminology of multidiversity (Allan et al. 2014 PNAS) and multifunctionality (Byrnes et al. 2014 Methods in Ecology and Evolution). More detailed descriptions of both multidiversity and multifunctionality are furthermore included in L168-181 and their calculations are additionally explained in the Supplementary Information. For

both indices we used the same mathematical framework, i.e. a threshold-based approach sensu Byrnes et al. 2014 Methods in Ecology and Evolution.

P2 L66 Can the genetic algorithm be described in plain English? If not, you might consider removing it from the high-level summary.

RESPONSE: We agree and removed the term from the Summary Paragraph.

P2 L69 “can only be reduced” seems too strong. Strong governance and regulation around spatial planning is one way that public policy can curb biodiversity loss and land use change instead of appealing to set the price right for the market to do the work.

RESPONSE: Thank you for this suggestion. We agree, and revised the sentence accordingly (“These findings suggest that, to reduce losses in biodiversity and ecosystem functioning, changes in economic incentive structures through well-designed policies are urgently needed”) (L70-72). Moreover, we added the importance of governance and regulation at the end of the Discussion (“Any approach will require law enforcement and the consideration of trade-offs between multifunctionality and profit in spatial planning to halt unsustainable land-use change and biodiversity loss in tropical lowlands.”) (L267-270).

Pg 4 L 186 You might briefly explain what pareto-optimum means for non-economists.

RESPONSE: We added a short explanation in L201-204: “In other words, the Pareto-frontier provided a set of multiple optimum landscape compositions, which cannot be further optimized (e.g., higher biodiversity) under the given constraints (i.e., the minimum expected profits per ha).” Since it is impossible to achieve maximal values for all the objectives such as biodiversity and ecosystem functioning indicators and profits from smallholder land-use simultaneously, we rely on pareto optimization. A solution is pareto optimal if the outcome cannot be improved for any objective such as biodiversity without deteriorating at least one of the other objectives such as profit. For our simulation, we set different minimum levels for expected profit.

Pg 4 L 196 How do these findings reconcile with other work by the authors that suggests palm oil is more profitable than rubber, and of course the revealed preference of those shifting?

RESPONSE: In general, our algorithm approach to identify optimized landscapes reconciles quite well with the fact that oil palm is overall more profitable than rubber – this is particularly pronounced when the focus is on multidiversity and high profit expectations, where the algorithm solutions indicate that economic profits are most efficiently (i.e. with the least costs in multidiversity) generated by including mainly oil palm in the conceptual landscape (in addition to rainforest for species conservation). However, we also find that to maintain high diversity within specific taxonomic groups as well as for high multifunctionality, landscapes composed by diverse mixture of different land-use types are often more appropriate. Thereby, the optimal landscape composition strongly varies depending on the target ecosystem functions. These ideal landscape situations of course do not match well with the actual current developments in land-use in Jambi Province – in fact, the preference of smallholders for oil palm leads in many parts to undesirable outcomes with

respect to ecosystem functioning (because the “real” landscape composition strongly differs from the theoretically “optimal” composition). This is why we highlight the need for more appropriate and targeted policies that create incentives for sustainable spatial landscape development, such as premium prices for certified jungle rubber production, which could help to maintain the share of this very valuable land-use system (in terms of ecosystem functions and biodiversity) in the landscape mosaic.

Pg 6 L 244 I think this study is incredibly valuable but it does not tell us what specific policy interventions will work best to manage the trade-offs. The authors have three options here: mention specific interventions, mention vague policy directions, or say nothing. Vague directions are weaker than being specific but you do not have the evidence in this paper to support being specific. I do not think you should be using studies from other contexts to justify advocating particular interventions, as we frankly still know little about what works and cannot generalize these past studies over. What this study does is help to highlight the primacy of economic drivers here and the necessity of policy design accounting for these (for example, through PES type approaches) or neutralizing them (through top-down regulations). If I were the authors, I would stop short of recommending (c.f., floating as options) any particular solution as you have not trialled any such things in the paper. Thus, this last discussion could perhaps be pared back. It is certainly the weakest part of the paper, in my view, even though I broadly agree with the positions staked and directions proposed.

RESPONSE: Thank you very much for your valuable comments. In light of your arguments and also the criticism by Reviewer #3 “The final discussion section around REDD and PES solutions also does not move our understanding forward, [...]”, we decided to significantly shorten our discussion of policy interventions.

Ryan Edwards

Reviewer #3 (Remarks to the Author):

The paper tackles a crucial question about trade-offs in biodiversity and livelihood profits for smallholders, and the best designs for mitigating trade-offs in these landscapes. The paper builds on other published studies by the research team at their Sumatra site analysing the same/similar data sets (e.g. Clough et al. Nature Comms 2016).

The focus of the paper on ecological-economic trade-offs is very important. The authors present a large amount of information from a smallholder landscape of 700 farms cultivating rubber or oil palm. They conclude that biodiversity conservation requires economic incentives to be changed in order to protect remaining areas of forest. This is an important finding, but not a new conclusion.

The focus on smallholders is important because these landscapes are generally less well-studied than industrial plantations – and these smallholder producers comprise a large proportion of agricultural landscapes and make large contributions to global rubber and oil palm production, particularly on Sumatra. The paper presents a substantial piece of work, reporting data from 14 taxa and 10 ecological functions from 700 smallholder farms.

The trade-offs the authors report occur because essentially two of the habitat types they study have high biodiversity/desirable function but no/low profit (degraded forest and jungle rubber), whereas the other two habitats are monocultures of rubber and oil palm which have low biodiversity/function but high profit. The genetic algorithm modelling confirms that selecting landscapes to maintain high multidiversity results in more forest, whilst maintaining high profit results in more oil palm. The solution for multifunctionality are less straightforward, with no consensus solution to maintain all functions. The authors then go on to suggest PES/REDD in order to provide enhanced livelihood benefits from forest.

This paper includes a huge amount of field data, carefully presented to address an important problem. The paper builds on other published studies from the team/study system. The authors should be congratulated on their field work efforts and synthesis. However, I am unclear what is conceptually novel in their study – we know from many studies (including the authors' works) that tropical forests are hyper diverse, such that land-use change for agriculture (esp. rubber and oil palm etc.) leads to declines. Hence the study arguably adds to the current knowledge on land-use change impacts and ecological-economic trade-offs, but provides no new understanding. It would have been helpful if the authors could have explained more clearly at the outset about the novelty of their work, and what new conclusions now arise compared with theirs and others' previous studies. I am not sure what the modelling of theoretical landscapes adds to the understanding of landscape change beyond that presented in the empirical data analyses (i.e. less forest = more profit but less diversity). I think this novelty should be much more clearly stated. The final discussion section around REDD and PES solutions also does not move our understanding forward, and REDD/PES arguments are known to rely on very high prices for carbon to match economic benefits from agriculture, much higher prices than current carbon markets. This point has been raised in many other studies.

So I think this is a very valuable data set, but suggest the authors take more care to explain what is already known (including in their study system), what is not, and how their study address new topics e.g. L80 – simply stating its 'poorly studied' is a weak justification for the study aims. Major restructuring of the paper to focus on novel investigations and findings would produce a much stronger and valuable paper.

RESPONSE: Thank you very much for your appreciation of our work and putting our study in the scientific context. We agree that the novel points of this study could be better highlighted in the introductory parts of our manuscript. We therefore revised the Introduction to highlight the multiple novel points that we address as compared to previous interdisciplinary work in our study system, in particular the work by Drescher et al. (2016), Phil Trans R Soc B and by Clough et al. (2016), Nat Comm.

Compared to these and other studies, we believe that this manuscript builds on four major novelties:

- 1) While we already know from previous studies in our study systems that smallholders switch from agroforestry to monoculture plantations, owing to the higher profitability of monoculture plantations per unit area, previous studies (Clough et al. 2016, Drescher et al. 2016) did not explicitly address the underlying relationship between economic profitability and ecological functions or biodiversity. Hence, it remained unclear whether these relationships are linear or non-linear, whether they vary among*

- the investigated taxonomic groups or ecological functions, and whether trade-offs are ubiquitous or also synergies exist. Our current manuscript addresses these important questions, which we now highlight in the Introduction in L89-91 and L104-106.*
- 2) In addition, by focussing on measures of multifunctionality and multidiversity, we investigate for the first time in our study system whether a) generalities regarding economic-ecological trade-offs emerge when assessing large suits of ecosystem functions or taxonomic groups, respectively, and b) whether the strength of trade-offs depends on the chosen threshold of the multifunctionality/-diversity index. The latter can be interpreted as a proxy of different management goals – the higher the threshold, the more stringent the expectation regarding the contribution of taxonomic groups or ecosystem functions to multidiversity or multifunctionality. This is highlighted in L106-109 and L168-181.*
 - 3) Also novel is the inclusion of many highly-diverse and ecologically important taxonomic groups (e.g. canopy ants, canopy parasitoid wasps, bats, butterflies, fungi) and critical ecosystem functions (e.g., soil greenhouse gas fluxes, plant transpiration) for which data were not available yet in previous works (Clough et al. 2016, Drescher et al. 2016). Including these taxonomic groups and ecosystem functions resulted in the most complete picture of how land-use decisions by smallholder farmers in the lowlands of Jambi shape biodiversity and ecosystem functioning in tropical ecosystems (included in L106-109). Such detailed and highly replicated interdisciplinary assessments are unique for tropical landscapes.*
 - 4) Finally, we aim to make first steps towards upscaling our findings at plot-level to the landscape scale by using a landscape optimization procedure to identify optimal landscape compositions that mitigate economic-ecological trade-offs for various profit expectations (now mentioned in L109-111). Although our approach is arguably still relatively simple (e.g., it does not consider effects of landscape configuration or other socioeconomic dimensions than economic profits), we believe it represents a first major step towards identifying landscape compositions that may better align with management that aims at meeting economic and ecological expectations in a more sustainable manner than currently done. Our results indicate that reconciling economic expectations with biodiversity conservation goals may necessitate different landscape management (e.g., landscapes dominated by forest and oil palm) than landscapes that target trade-offs between economic profits and ecosystem functions (whereby the optimal landscape composition strongly depends on the targeted ecosystem function, i.e. including mixes of rubber, jungle rubber and oil palm). Nonetheless, our results indicate that, even for optimally composed landscapes, trade-offs between economic and ecological functions will always exist. Hence, the exercise demonstrates (in a simplified manner) that solely striving for higher economic profits will always necessarily compromise biodiversity and ecological functions, and that policies that address these unsustainable developments are therefore urgently needed.*

Moreover, in light of your arguments and those by Reviewer #2 (“If I were the authors, I would stop short of recommending (c.f., floating as options) any particular solution as you have not trialled any such things in the paper.”), we significantly shortened our discussion of policy suggestions at the end of the manuscript, as these were relatively vague and did not really built on the empirical analysis.

----- END